# A unifying criterion of the boiling crisis

Limiao Zhang[1,2], Chi Wang[1,3], Guanyu Su[1,4], Artyom Kossolapov[1], Gustavo Matana Aguiar[1], Jee Hyun Seong[1,5], Florian Chavagnat[1], Bren Phillips[1], Md Mahamudur Rahman[1,6] & Matteo Bucci[1] ✉

We reveal and justify, both theoretically and experimentally, the existence of a unifying criterion of the boiling crisis. This criterion emerges from an instability in the near-wall interactions of bubbles, which can be described as a percolation process driven by three fundamental boiling parameters: nucleation site density, average bubble footprint radius and product of average bubble growth time and detachment frequency. Our analysis suggests that the boiling crisis occurs on a well-defined critical surface in the multidimensional space of these parameters. Our experiments confirm the existence of this unifying criterion for a wide variety of boiling surface geometries and textures, two boiling regimes (pool and flow boiling) and two fluids (water and liquid nitrogen). This criterion constitutes a simple mechanistic rule to predict the boiling crisis, also providing a guiding principle for designing boiling surfaces that would maximize the nucleate boiling performance.

The importance of boiling cannot be overstated. Boiling is crucial to the operation, efficiency and safety of technologies (e.g., electric energy production, sterilization, water desalination, power electronics, medical diagnosis and therapy, fission and fusion energy, high-performance computing, space exploration and colonization) that are critical to the present and future of humankind.

Nucleate boiling (i.e., boiling via the nucleation of bubbles at a heated surface) is an effective heat removal process. The amount of energy required to transform liquid into vapor and grow a bubble is typically large compared to the energy necessary to increase the liquid temperature, e.g., from ambient temperature to its boiling point[1,2]. This energy is provided by the surface from which heat is to be removed, e.g., the cladding of a nuclear reactor fuel rod, where bubbles nucleate at discrete locations called nucleation sites. The heat flux that can be removed by nucleate boiling depends on nucleation sites area density (simply called nucleation site density), bubble size and release frequency. When, starting from a stable operating condition (see Fig. 1a), the heat flux to be removed from the heated surface is increased, the process finds a new point of equilibrium by self-adjusting these parameters (e.g., by increasing the nucleation site density). However, at high heat fluxes, this process may become unstable. When that happens, the heated surface suddenly gets

covered by a stable vapor film (see Fig. 1b). This instability, known as a boiling crisis, is a key operational limit in many systems, e.g., nuclear reactors, as this vapor film has poor heat transfer properties[3] (see Fig. 1d). Thus, when the boiling crisis occurs, the surface temperature (see Fig. 1e) may suddenly increase up to the point where the surface burns out (see Fig. 1c) or even melts, causing the system's failure.

The boiling crisis is a century-old scientific problem. For years, it has been viewed as the outcome of a hydrodynamic instability occurring far from the heated surface[4]. When the release frequency of bubbles from nucleation sites is too high, bubbles merge to form vapor columns. The flow of vapor within these columns increases with the heat flux. However, there is a critical vapor velocity, above which these vapor columns deform due to Helmholtz instabilities and merge, obstructing the flow of liquid towards the heated surface, which eventually dries out. Antithetical descriptions consider the boiling crisis as a near-wall instability related to the characteristic of the nucleate boiling process at the heated surface. For instance, it has been suggested that the boiling crisis is the consequence of a mechanical unbalance of the liquid-vapor-solid contact line, triggered by the recoil forces that generate when the near-wall liquid vaporizes[5–8]. In some models, the boiling crisis is thought of as a consequence of a critical packing of bubble on the heated surface, due to an increase of the

[1]Department of Nuclear Science and Engineering, Massachusetts Institute of Technology, Cambridge, MA 02139, USA. [2]Information Materials and Intelligent Sensing Laboratory of Anhui Province, Anhui University, Hefei, China. [3]Department of Mechanical Science and Engineering, University of Illinois at Urbana-Champaign, Urbana, IL 61801, USA. [4]Department of Nuclear Engineering, University of California, Berkeley, CA 94709, USA. [5]Los Alamos National Laboratory, Los Alamos, NM 87545, USA. [6]Department of Mechanical Engineering, University of Texas at El Paso, El Paso, TX 79968, USA. ✉e-mail: mbucci@mit.edu

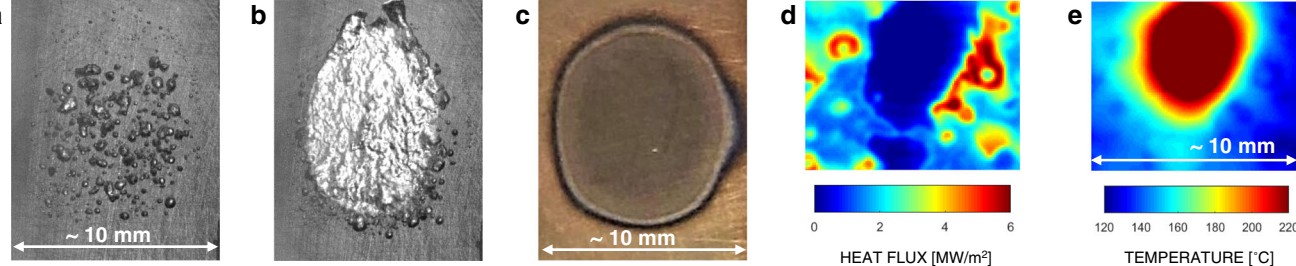

**Fig. 1 | The boiling crisis. a** High-speed video image of the nucleate boiling process. **b** High-speed video image of the film boiling process (on the same surface as **a**). a and b show examples obtained in flow boiling conditions on a vertical surface. **c** Picture of a burnt-out surface which experienced the boiling crisis. This burnout may be followed by the melting of the surface. **d** Heat flux distribution on a boiling surface experiencing film boiling. **e** Temperature distribution on a boiling surface experiencing film boiling. **d** and **e** are simultaneous measurements obtained using infrared thermometry and have the same scale bar. The large patch with poor heat transfer properties (zero heat flux in **d**) and high temperature (hot spot in **e**) indicate the presence of a stable vapor film covering the heated surface.

nucleation site density[9]. In some other models, the key to predicting the boiling crisis is the bubble growth time, i.e., the time a bubble takes to grow and leave the heated surface. The bubble growth time determines the rise of surface temperature during the bubble life cycle, which is the critical parameter in some models[10–12]. Decreasing the bubble growth time is beneficial, as it reduces the temperature rise. It is also beneficial because it increases the bubble release frequency and, consequently, the heat flux that can be removed by the nucleate boiling process[13]. However different, all these models have a common denominator: they all assume that the boiling crisis can be predicted as the outcome of a phenomenon triggered by a single critical characteristic quantity, e.g., the vapor velocity in the vapor columns or the nucleation site density or the growth time. As models often come from empirical observations of the boiling process in different operating conditions or on surfaces with different physicochemical properties, the diversity and specificity of these models may indicate that there is not a unique triggering mechanism of the boiling crisis, or that we still have not found the central idea that reconciles all these observations.

Here, we prove the existence of a criterion that captures the boiling crisis over a broad range of surface and operating conditions by unifying into a single paradigm the synergistic and intertwined effect of the length and time scales involved in the boiling process (i.e., nucleation site density, average bubble footprint size, growth time and departure frequency).

## Results

We have recently discovered the signature of the boiling crisis in the probability density function of the bubble footprint areas on the heated surface[14,15]. This discovery has been possible through experiments (see examples of subcooled flow boiling results on a rough zirconium surface in Fig. 2) featuring high-resolution optical diagnostics (e.g., infrared thermometry[16] and phase detection[17]) developed in-house (see *Methods*), which enable detection of bubble footprints and other boiling parameters (e.g., bubble footprint area distributions, circular-equivalent footprint radius distribution of non-interacting bubbles, bubble growth time and detachment frequency, and nucleation site density). We observed that, for low heat fluxes, when there are only a few bubbles growing on the boiling surface (e.g., see Fig. 2b, c at 0.79 MW/m² in the example of the figure), the bubbles do not interact with each other, and their footprint area distribution is a damped function with a proper mean value and standard deviation (i.e., discrete bubbles have, as expected, a proper length scale). As the heat flux is increased and more bubbles grow on the surface, they start to interact occasionally forming larger vapor patches (e.g., see Fig. 2c at 2.36 MW/m² in the example of the figure). Upon occurrence of the boiling crisis (see Fig. 2b, c at 3.48 MW/m² in the example of the figure), this distribution follows a scale-free trend, i.e., a power-law distribution with a negative exponent < 3. The occurrence of a scale-free

distribution has profound implications. Its standard deviation is infinite. As such, it is not possible to define this distribution by means of a dominant, central value, such as for a normal distribution. Instead, all scales are equally important. This observation inspired us to recognize the importance of seldom, large fluctuations of the bubble footprint area in triggering the boiling crisis and to realize that, to fully understand and predict the boiling crisis, one cannot only focus on the behavior of a single bubble, but needs to look at how bubbles behave collectively, at all scales. Describing their interaction correctly and predicting these bubble footprint fluctuations is the key to predicting the boiling crisis.

We have captured this behavior by modeling the bubble interaction as a stochastic continuum percolation process (see Fig. 2d and Methods or Refs. [14], [15] for the details of the model), driven by three fundamental boiling parameters: nucleation site density $N''$, average footprint radius $R$ of non-interacting bubbles and product of average bubble growth time $t_g$ and detachment frequency $f$, which expresses the probability to have a bubble growing at a nucleation site (note that this probability is between 0 and 1, as $f = 1/(t_g + t_w)$, where $t_w$ is the time elapsed from the departure of a bubble until the nucleation of a new one from the same nucleation site). This model predicts how the experimental bubble footprint area distribution changes with the heat flux through the aforesaid (measured) boiling parameters (see Fig. 2e). It confirms that, at the boiling crisis, the bubble footprint area distribution is scale-free. Importantly, it reveals that the boiling crisis (i.e., the power-law distribution) coincides with a bifurcation. Figure 2f shows how the area of the most likely giant (G) and second giant (SG) bubble cluster (i.e., the bubble clusters with the largest and second largest footprint area, respectively) changes with the heat flux. In nucleate boiling (i.e., for heat fluxes below 3.48 MW/m² for in the example of Fig. 2f), the size of these clusters is comparable and grows monotonically. However, if, starting from the critical distribution (i.e., 3.48 MW/m² in the example of Fig. 2), one increases $N''$, or the average footprint radius $R$, or the product $f t_g$, i.e., if one increases the heat flux, the model intrinsically predicts that all the bubble clusters suddenly merge together into a unique giant vapor patch (see Fig. 2f). This behavior is confirmed by the prediction of a supercritical bubble footprint area distribution (see Fig. 2e). In other words, the model predicts a sudden transition from a nucleate boiling regime to a stable vapor film, called film boiling regime (see Fig. 2b). This analysis suggests that the boiling crisis is triggered by an instability in the bubble interaction process, which occurs for a critical combination of $N''$, $R$, and $f t_g$, i.e., a critical triplet.

Ideally, critical, scale-free distributions can be obtained with infinite combinations of these three parameters, i.e., there is an infinite number of critical triplets. Such triplets can be identified using the stochastic model. Consider the case where average footprint radius of non-interacting bubbles, growth time and departure frequency are fixed, and the nucleation site density progressively increases starting

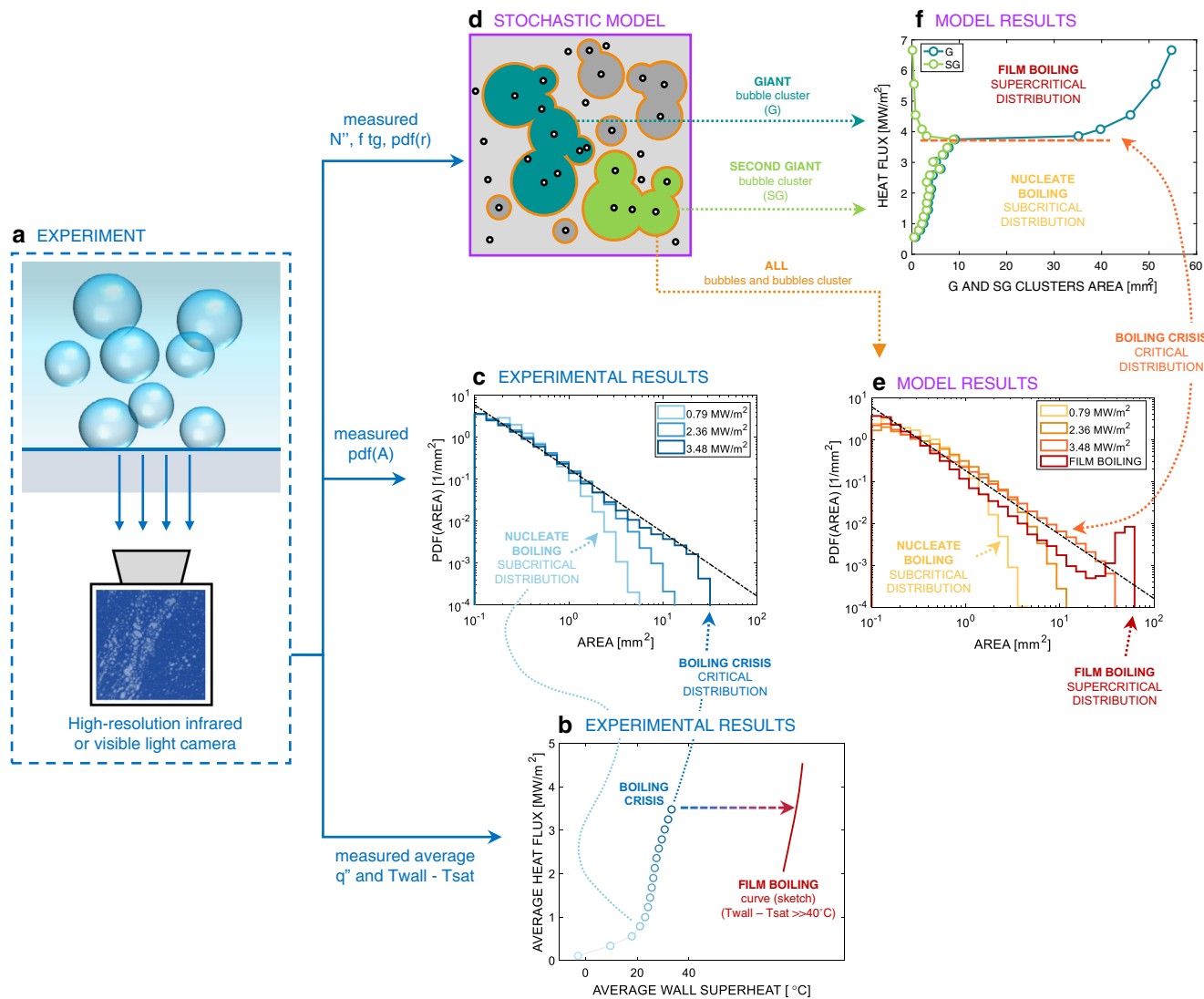

**Fig. 2 | Synopsys of the theoretical and experimental methodology and sample results for subcooled flow boiling experiments run on a rough zirconium surface (see boundary conditions in Table 2). a** Sketch of the experimental apparatus and high-resolution camera for infrared thermometry or phase-detection measurements. **b** Experimental boiling curve. **c** Experimental bubble footprint area distributions. **d** Example of the stochastic bubble interaction model output. **e** Bubble footprint distributions predicted by the stochastic model. **f** Evolution of the giant (G) and second giant (SG) bubble clusters with heat flux predicted by the model, showing that a power law bubble footprint area distribution coincides with a bifurcation of the G and SG clusters area.

from a value for which bubbles are sparse and do not interact with each other. In these conditions, the bubble footprint area distribution follows a damped function (i.e., the footprint area distribution of non-interacting bubbles). As the nucleation site increases, this distribution changes, until for a certain value of the nucleation site density, this distribution becomes scale free. Starting from this distribution, a further addition of nucleation sites would result in the presence of a unique vapor cluster on the boiling surface (i.e., the film boiling regime). The bifurcation in the G and SG cluster size (as same observed in Fig. 2f) can be used to identify the critical triplet. This modeling exercise can be repeated by changing the initial $R$, or $f t_g$, or, in general, by keeping any of the parameters fixed except one. By doing so, we could identify a wide number of critical triplets. Notably, they are described by a simple non-dimensional formula (see Fig. 3a):

$$N^{''}\pi R^2 f t_g = C \tag{1}$$

Here, $C$ depends on the non-dimensional boiling area $A_h/\pi R^2$ (where $A_h$ is the area of the boiling surface). It increases from 0.95, for a nondimensional area of 10, to an asymptotic value of 1.15 for a non-

dimensional area of 10000 or larger (see *Methods*). In the typical range of non-dimensional boiling area of literature experiments, including ours (mostly between 50 and 300), the average $C$ value is 1.03 with a standard deviation $\sigma_C = 0.06$. Note that $N^{''}\pi R^2$ is the area fraction that would be covered by bubble, if they were not interacting, and $f t_g$ is the probability that a bubble exists. We could be tempted to think that the boiling crisis (i.e., $N^{''}\pi R^2 f t_g \sim 1$) occurs when bubbles cover, on average, the entire heated surface—a mental picture that recall the image drawn in the early work of Rohsenow and Griffith[9]. However, Eq. (1) does not imply that the entire surface is simultaneously covered by bubbles. In fact, even in critical conditions, only fraction of the surface (typically between 40% and 60%) is covered by bubbles simultaneously.

## Discussion

The theoretical scaling expressed by Eq. (1) is a criterion of the boiling crisis. It defines the criticality boundary between a stable and efficient nucleate boiling regime, and a stable but inefficient film boiling regime. To verify this criterion, we conducted experiments in a broad range of operating and surface texture conditions. These experiments feature high resolution optical techniques to image the boiling surface

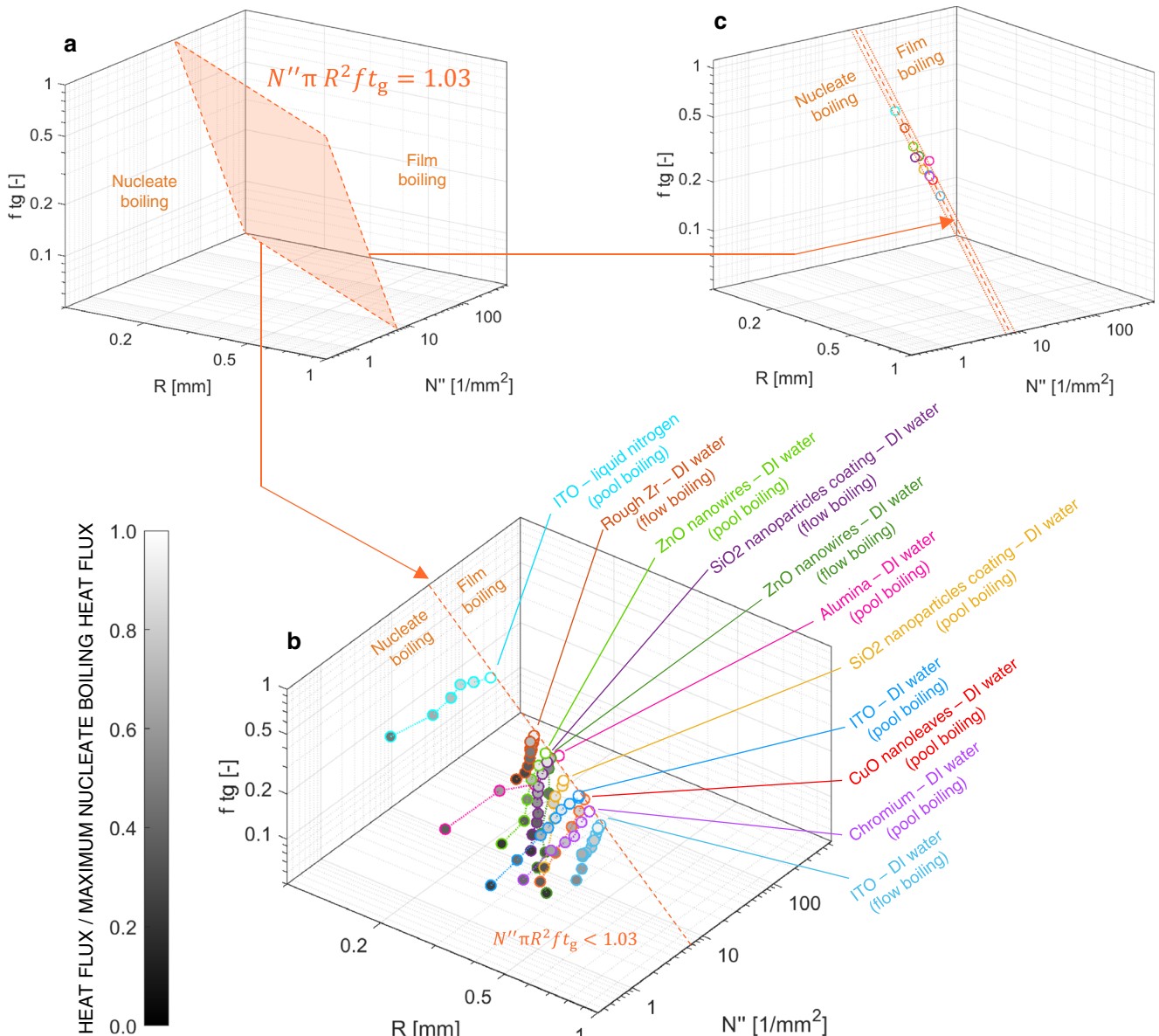

**Fig. 3 | The unifying criterion of the boiling crisis. a** Critical surface determined using the stochastic model. **b** Representation of the experimental boiling triplets $(N'', R, f t_g)$ for several boiling surfaces and operating conditions (Source data are provided as a Source Data file). The dot grayscale is proportional to the ratio between the heat flux and the maximum heat flux that each surface (represented by different line colors) can remove in nucleate boiling. White dots indicate the experimental boiling crisis and lie on the theoretical critical surface $N'' \pi R^2 f t_g = 1.03$. **c** Simplified view of the critical surface (dashed line) and the experimental boiling triplets $(N'', R, f t_g)$ at the boiling crisis for all surfaces and operating conditions. Solid lines represent the standard deviation on the theoretical critical constant (see Methods). An illustration of the effects of measurement uncertainties is shown in the Methods section.

and measure the time-dependent distributions of the temperature and heat flux, or the phase (liquid or vapor) distributions in contact with the boiling surface (see Methods). From these measurements, we can quantify the number of nucleation sites (i.e., the nucleation site density, $N''$), the circular-equivalent footprint radius distribution of discrete, non-interacting bubbles, $p(r)$, as well as the bubble growth time $t_g$ and the bubble departure frequency $f$ on any kinds of surface we tested (see Methods for the details). We investigated the pool boiling of saturated water at atmospheric pressure on nano-smooth indium tin oxide (ITO), nano-smooth chromium, nano-smooth alumina, copper oxides nano-leaves, zinc-oxide nanowires, and coatings made of silicon dioxide nanoparticles. We also investigated the pool boiling of liquid nitrogen at atmospheric pressure on ITO, and the flow boiling of water at atmospheric pressure with 10 K of subcooling on ITO, zinc-oxide nanowires, coatings made of silicon dioxide nanoparticles, and a rough

Zirconium surface. These surfaces and operating conditions (summarized in Tables 1 and 2) were selected to cover a broad range of boiling triplets (at the boiling crisis, the $f t_g$ values range between 0.28 and 0.53, the $R$ values between 0.17 and 0.61 mm, and the $N''$ values between 222 and 2195 sites per cm²). Figure 3b shows values of the experimental boiling triplets for all the surfaces and operating conditions we tested. The line color of each dot indicates a specific surface and operating conditions. Instead, the dot grayscale intensity for each series of data is proportional to the ratio between the heat flux and the maximum nucleate boiling heat flux that could be removed by the selected surface and operating condition (also known as critical heat flux or CHF). In general, experimental triplets are located at the left of the critical surface, i.e., their $N'' \pi R^2 f t_g$ product is <1.03. However, the last (and brightest) point of these experimental three-dimensional boiling curves (i.e., the point at which the boiling crisis occurs) lies on

**Table 1 | Data of heat flux, wall superheat, average bubble footprint radius, *R*, product of bubble growth time and departure frequency, *ft*$_g$, and nucleation site density, *N*$''$ , for all surfaces and operating conditions tested in pool boiling**

| | No. | Surface Fluid | Heat flux [kW/m²] | Wall superheat [°C] | *R* [mm] | *ft*$_g$[−] | *N*''[1/cm²] |
|---|---|---|---|---|---|---|---|
| Pool boiling | 1 | ITO DI water | 300 | 19.5 | 0.38 | 0.12 | 70 |
| | | | 500 | 20.7 | 0.42 | 0.16 | 110 |
| | | | 800 | 24.1 | 0.44 | 0.20 | 193 |
| | | | 900 | 25.6 | 0.45 | 0.22 | 223 |
| | | | 1030 | 26.3 | 0.47 | 0.26 | 233 |
| | | | 1080 | 25.9 | 0.48 | 0.29 | 261 |
| | | | 1100 | 25.9 | 0.50 | 0.33 | 275 |
| | | | 1150 | 26.5 | 0.50 | 0.33 | 370 |
| | | | 1180 | 26.3 | 0.50 | 0.33 | 377 |
| | 2 | ZnO nanowiresDI water | 330 | 15.6 | 0.39 | 0.21 | 93 |
| | | | 550 | 17.3 | 0.36 | 0.19 | 271 |
| | | | 770 | 18.8 | 0.33 | 0.21 | 424 |
| | | | 1090 | 20.9 | 0.33 | 0.28 | 481 |
| | | | 1360 | 22.7 | 0.34 | 0.34 | 526 |
| | | | 1470 | 23.6 | 0.36 | 0.42 | 563 |
| | 3 | CuO nanoleaves DI water | 720 | 23.8 | 0.51 | 0.15 | 95 |
| | | | 1020 | 27.3 | 0.49 | 0.17 | 199 |
| | | | 1310 | 31.0 | 0.48 | 0.20 | 370 |
| | | | 1580 | 35.3 | 0.48 | 0.24 | 461 |
| | | | 1790 | 37.6 | 0.49 | 0.29 | 480 |
| | | | 1920 | 38.4 | 0.48 | 0.30 | 469 |
| | 4 | SiO2 nano-particles DI water | 760 | 24.6 | 0.47 | 0.14 | 158 |
| | | | 1410 | 34.6 | 0.40 | 0.19 | 435 |
| | | | 1810 | 41.7 | 0.39 | 0.23 | 465 |
| | | | 1970 | 44.7 | 0.40 | 0.25 | 470 |
| | | | 2100 | 48.1 | 0.42 | 0.30 | 496 |
| | | | 2220 | 48.7 | 0.42 | 0.32 | 509 |
| | 5 | Chromium DI water | 410 | 24.9 | 0.55 | 0.22 | 41 |
| | | | 630 | 26.7 | 0.59 | 0.29 | 76 |
| | | | 750 | 28.3 | 0.59 | 0.29 | 113 |
| | | | 860 | 30.9 | 0.59 | 0.28 | 161 |
| | | | 920 | 32.7 | 0.60 | 0.33 | 188 |
| | | | 980 | 35.6 | 0.61 | 0.37 | 222 |
| | 6 | Alumina DI water | 650 | 24.0 | 0.28 | 0.22 | 65 |
| | | | 1040 | 29.3 | 0.29 | 0.25 | 292 |
| | | | 1330 | 32.8 | 0.33 | 0.24 | 585 |
| | | | 1570 | 36.5 | 0.33 | 0.27 | 906 |
| | | | 1710 | 40.0 | 0.34 | 0.32 | 1026 |
| | 7 | ITO liquid nitro-gen (78 K) | 90 | 5.4 | 0.11 | 0.18 | 677 |
| | | | 130 | 6.2 | 0.13 | 0.25 | 1112 |
| | | | 150 | 6.4 | 0.14 | 0.33 | 1364 |
| | | | 160 | 6.7 | 0.15 | 0.39 | 1615 |
| | | | 180 | 7.0 | 0.16 | 0.42 | 1760 |
| | | | 200 | 7.2 | 0.17 | 0.45 | 2195 |

These tests were performed in saturation conditions at ambient pressure. Source data are provided as a Source Data file.

top of the critical boundary defined by Eq. (1) (see Fig. 3c for a clearer representation and the Methods section for an illustration of the effects of measurement uncertainties). In other words, the boiling crisis happens when the non-dimensional boiling crisis number

$N''\pi R^2 ft_g$ calculated with the measured boiling triplet becomes 1.03, no matter the boiling surface or operating condition. We emphasize that, while we did not test hydrophobic surfaces with static contact angles much >90°, our paradigm contemplates all surface conditions

**Table 2 | Data of heat flux, wall superheat, average bubble footprint radius, $R$, product of bubble growth time and departure frequency, $ft_g$, and nucleation site density, $N''$, for all surfaces and operating conditions tested in flow boiling**

| | No. | Surface Fluid | Heat flux [kW/m²] | Wall superheat [°C] | $R$ [mm] | $ft_g$ [-] | $N''$ [1/cm²] |
|---|---|---|---|---|---|---|---|
| Flow boiling | 8 | Rough Zr DI water (Example of Fig. 2) | 790 | 21.1 | 0.33 | 0.37 | 131 |
| | | | 1000 | 23.1 | 0.35 | 0.36 | 248 |
| | | | 1230 | 24.1 | 0.35 | 0.38 | 299 |
| | | | 1450 | 25.0 | 0.35 | 0.41 | 330 |
| | | | 1680 | 25.6 | 0.35 | 0.45 | 350 |
| | | | 1900 | 26.2 | 0.35 | 0.48 | 376 |
| | | | 2130 | 26.8 | 0.35 | 0.48 | 404 |
| | | | 2360 | 27.4 | 0.34 | 0.48 | 405 |
| | | | 2580 | 28.3 | 0.34 | 0.50 | 438 |
| | | | 2790 | 29.5 | 0.34 | 0.52 | 438 |
| | | | 3020 | 30.6 | 0.34 | 0.49 | 475 |
| | | | 3250 | 31.9 | 0.34 | 0.51 | 501 |
| | | | 3480 | 33.2 | 0.33 | 0.53 | 514 |
| | 9 | SiO2 nanoparticles DI water | 970 | 33.5 | 0.31 | 0.02 | 164 |
| | | | 1310 | 36.6 | 0.31 | 0.04 | 308 |
| | | | 1730 | 39.2 | 0.34 | 0.10 | 408 |
| | | | 2150 | 40.8 | 0.34 | 0.13 | 433 |
| | | | 2550 | 42.7 | 0.34 | 0.15 | 489 |
| | | | 2960 | 45.5 | 0.35 | 0.17 | 509 |
| | | | 3380 | 49.2 | 0.34 | 0.19 | 519 |
| | | | 3760 | 53.4 | 0.34 | 0.22 | 533 |
| | | | 3930 | 55.9 | 0.34 | 0.24 | 553 |
| | | | 4280 | 61.2 | 0.35 | 0.29 | 582 |
| | | | 4430 | 63.1 | 0.35 | 0.36 | 602 |
| | 10 | ZnO nanowires DI water | 970 | 31.4 | 0.21 | 0.00 | 168 |
| | | | 1200 | 35.6 | 0.39 | 0.02 | 311 |
| | | | 1440 | 36.6 | 0.39 | 0.06 | 375 |
| | | | 1670 | 37.9 | 0.35 | 0.08 | 464 |
| | | | 1910 | 38.1 | 0.35 | 0.09 | 591 |
| | | | 2380 | 39.2 | 0.34 | 0.11 | 622 |
| | | | 3330 | 44.6 | 0.35 | 0.21 | 676 |
| | | | 4060 | 49.4 | 0.35 | 0.30 | 710 |
| | | | 4410 | 53.3 | 0.35 | 0.35 | 704 |
| | 11 | ITO DI water | 2030 | 35.7 | 0.68 | 0.20 | 87 |
| | | | 2320 | 36.5 | 0.67 | 0.21 | 114 |
| | | | 2430 | 36.7 | 0.65 | 0.24 | 131 |
| | | | 2550 | 37.1 | 0.65 | 0.25 | 130 |
| | | | 2670 | 37.7 | 0.64 | 0.23 | 146 |
| | | | 2780 | 38.1 | 0.64 | 0.23 | 165 |
| | | | 2900 | 38.8 | 0.63 | 0.23 | 198 |
| | | | 3010 | 39.4 | 0.63 | 0.25 | 219 |
| | | | 3130 | 40.1 | 0.63 | 0.24 | 218 |
| | | | 3250 | 40.8 | 0.62 | 0.25 | 242 |
| | | | 3370 | 41.6 | 0.62 | 0.26 | 255 |
| | | | 3470 | 42.4 | 0.62 | 0.27 | 276 |
| | | | 3590 | 43.4 | 0.62 | 0.27 | 273 |
| | | | 3770 | 44.6 | 0.62 | 0.28 | 300 |

These tests were performed with a mass flux of 1000 kg/m2s and a subcooling of 10 K at atmospheric pressure. Source data are provided as a Source Data file.

and, possibly, heater geometry effects. For instance, on hydrophobic surface, depending on nucleation sites size distribution and nucleation temperature, the boiling crisis may occur with a large number of small bubbles (i.e., with a small footprint) and a short wait time (i.e., a high $ft_g$)[18]. In some other cases, the bubble departure diameter on hydrophobic surfaces may be significantly higher than on hydrophilic surfaces[19], and the boiling crisis may happen with a combination of large bubble footprint radius and smaller nucleation site density (and likely higher $ft_g$ product). However, while experiments on superhydrophobic surfaces and considering other effects (e.g., heater size and lateral confinement) may be run in the future, the critical condition expressed by Eq. (1) should not change.

Last but not least, while Eq. (1) contemplates any surface conditions and bubble dynamics, it also yields the same scaling of the critical heat flux (CHF) with the physical properties of the fluid (and operating conditions) predicted by the Kutateladze-Zuber correlation[4] (see Methods). At the same time, our idea corroborates the hypothesis that the boiling crisis is a near-wall phenomenon, as we only used information related to the dynamic of bubbles at the heating surface. Interestingly, the concurrence of these observations suggest that our idea can reconcile far-field and near-wall views of the boiling crisis.

In summary, we prove with a theoretical and experimental investigation the existence of a unifying scaling criterion of the boiling crisis. This non-dimensional criterion was postulated through a stochastic continuum percolation model of the bubble interaction process, which captures large-scale bubble cluster fluctuations and predicts the boiling crisis as an instability of the bubble interaction process. Experimental data for a variety of surfaces, operating conditions and fluids confirm this theoretical unifying scaling criterion. We emphasize that, in this context, the word unifying is used because this criterion combines and captures the synergistic and intertwined effect of the length and time scales involved in the boiling process, i.e., nucleation site density, bubble size, growth time and departure frequency.

Equation (1) provides a simple mechanistic criterion to model the boiling crisis in advanced two-phase heat transfer computational tools used for the design of industrial and scientific systems, where it can be used in combination with heat flux partitioning approaches to predict the critical heat flux[15]. It also constitutes a guiding principle for designing boiling surfaces that would maximize the nucleate boiling performance, e.g., by engineering surface features that maximize the heat flux that can be removed by critical triplets.

## Methods

### Experimental methodology

We conduct pool and flow boiling experiments using special heaters, enabling the use of high-speed infrared (IR) thermometry to measure the time-dependent infrared radiation distribution emitted by the boiling surface. This technique is used in all the experiments with de-ionized (DI) water. Instead, we run experiments with liquid nitrogen (LN2) using a different optical technique called phase detection.

**Experiments with water.** Boiling experiments with water are conducted using special infrared heaters, designed and built in-house. They consist of a 1 mm thick sapphire substrate ($20 \times 20$ mm$^2$), coated on one side with a thin electrically conductive film, which is the actual Joule heating element (see Fig. 4c). This film can be made of Indium Tin Oxide (ITO) or metals, and it is less than a micron thick. It is connected to a power supply with two silver or gold pads with negligible electrical resistance, which define the active heating area (nominally $10 \times 10$ mm$^2$). This heater is installed at the bottom of our pool boiling facility or on the side wall of our flow loop, with the thin electrically conductive film in contact with water. We emphasize that the thermal response of the heaters used in this study is not determined by the thin heating layer, but rather by the underlying 1 mm thick sapphire substrate, which has a large thermal capacity. Importantly, sapphire has thermal properties (e.g., diffusivity and effusivity) very similar to stainless steel, Inconel and zircaloy, which are materials of interest for commercial boiling applications, e.g., in nuclear reactor fuel claddings.

The pool boiling facility (see Fig. 4a) consists of a concentric double cylinder structure, forming two separate fluid cells. The working fluid, DI water, is in the inner cell. A temperature-controlled oil is circulated throughout the outer cell, which is surrounded by an insulating material to minimize heat losses to the environment (this insulating material is not shown in the sketch of Fig. 4a). The external cell works as an isothermal bath and is used to bring the DI water to and maintain it at the desired temperature. The temperature of the two fluids (oil and DI water) is monitored using T-type thermocouples installed in both cells.

The free surface of the water is 15 cm above the heater. The latter is accommodated over a ceramic support structure sitting on the bottom of the inner cell. The inner cell of the apparatus is abundantly rinsed with DI water before the beginning of each experiment. Then, the heater is installed, and the cell is filled with DI water at room temperature. At this point, the temperature of the oil in the outer bath is gradually increased until the temperature of the DI water in the inner cell reaches saturation temperature. As the saturation temperature is reached, we leave the apparatus to thermalize for ~30 min. During this phase, we turn the heater power on for ~10 min to induce nucleate boiling and degas the heater surface as well. All the pool boiling experiments presented in this paper are conducted at atmospheric pressure.

The flow boiling facility (see Fig. 4b) consists of a stainless steel 316 L test section with a 3 cm × 1 cm flow channel running the length of the structure. It connects to an entrance region (not shown in Fig. 4b) over 60 hydraulic diameters long to establish fully-developed turbulent flow at the position of the heater. The main body of the test section consists of four sides. Three are used for quartz windows to provide optical access; the fourth wall contains a ceramic cartridge used to hold the IR heater perfectly flushed with the channel walls. The facility is equipped with variable frequency pump, flow meter, temperature and pressure instrumentation, preheater, chiller, accumulator, and a fill-and-drain tank. Filtering and dissolved oxygen monitoring is accomplished via a secondary loop used during the initial stages of testing. A pump provides the head required for the desired mass flux, i.e., 1000 kg/m$^2$s in these experiments. The bulk temperature of the fluid is controlled by adjusting the power of the preheater and the secondary flow in the chiller. The experiments discussed in this paper are run at atmospheric pressure with slightly subcooled water at 90 °C.

As we circulate current through the electrically conductive film of the heater, it releases the heat necessary to warm up the sapphire substrate and boil the water by Joule effect. Importantly, this electrically conductive film has negligible thermal capacity and thermal resistance. Thus, its temperature coincides with the temperature at the interface between the solid and the fluid, i.e., the boiling surface temperature. Noteworthy, while sapphire is quasi-transparent to infrared radiation in the 3–5 μm range, this film is perfectly IR opaque. It emits time-dependent and space-dependent radiation proportional to its temperature. We use an infrared camera (IRC806HS) to record the radiation emitted by the entire heater (i.e., nominally $10 \times 10$ mm$^2$) at a temporal resolution of 2500 frames per second (i.e., with a time step of 0.4 ms) and a pixel resolution of 115 μm. We have developed a calibration technique to convert this time-dependent radiation distribution into temperature and heat flux. The quasi-transparent (but not perfectly transparent) nature of the sapphire makes this process complicated. Sapphire partially absorbs and emits radiation with a wavelength >4 μm, contaminating the radiation signal emitted by the thin heating film, which is the one we need to measure. Reflections of the background radiation further contaminate this signal. The radiation emitted by the thin heating film is found through the solution of an inverse problem coupling optical radiation and conduction heat transfer in the substrate. For more details, we direct the reader to our previous work[16]. Using this calibration technique, we obtain the time-dependent distributions of boiling surface temperature, $T_w(x, y, t)$, and heat flux from the thin heating film to the sapphire substrate, $q_s''(x, y, t)$. The time-dependent distribution of the heat flux to water, $q_w''(x, y, t)$, is obtained as the difference between the Joule heating superficial power density and the heat flux to sapphire. Precisely, the Joule heating superficial power density, $q_h''$, is measured as the product of the current circulating through the heater, $I$, and the voltage drop across the silver pads, $V$, divided by the actual heating area, $A_h$.

$$q_h'' = \frac{V \cdot I}{A_h}. \tag{2}$$

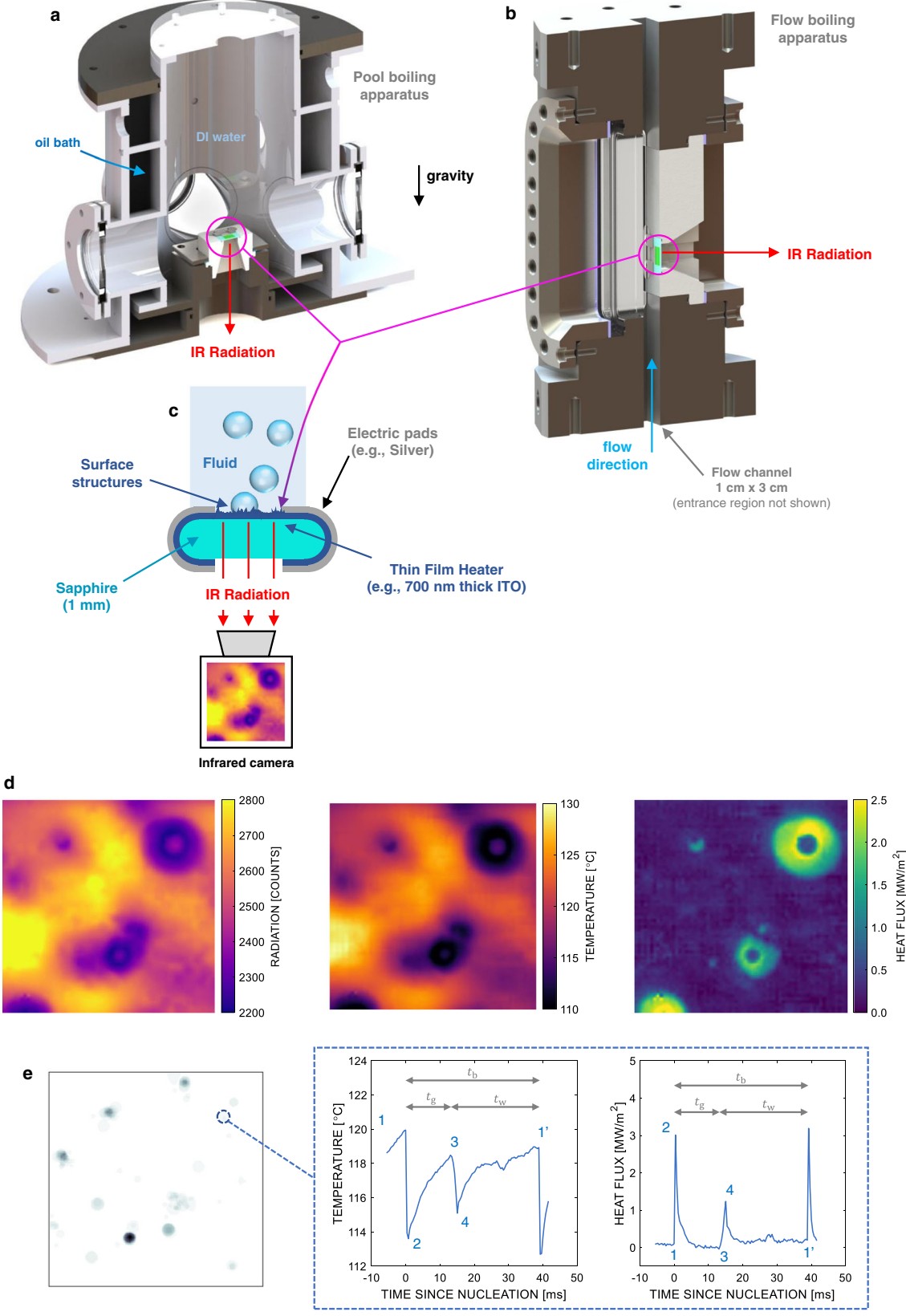

**Fig. 4 | Test sections used for water experiments with infrared heaters and sample results. a** Cross sectional view of the pool boiling apparatus used for boiling experiments with water. **b** Cross sectional view of the flow boiling apparatus used for boiling experiments with water. **c** Sketch of the infrared heater (not to scale). **d** Examples of instantaneous infrared radiation distribution (left) and the resulting temperature (middle) and heat flux distributions (right). **e** Nucleation probability map (in arbitrary units) with the identified nucleation sites (left), and example of temperature (middle) and heat flux (right) time-evolution at a selected nucleation site throughout a bubble life cycle.

The time-dependent distribution of the heat flux to water is eventually obtained as

$$q_w''(x,y,t) = q_h'' - q_s''(x,y,t). \tag{3}$$

During the experiments, the heat flux is increased in steps, until the boiling crisis occurs. For each heat flux step, we record the infrared radiation emitted by the heater for 2 s. The time-dependent infrared radiation distributions can be post-processed to obtain the time-dependent temperature and heat flux distributions according to the technique introduced before and discussed in detail in our previous work[16]. Figure 4d shows an example of instantaneous infrared radiation and the associated temperature and heat flux distributions.

These time-dependent distributions can be analyzed to extract fundamental boiling parameters, e.g., nucleation site density, bubble growth time, bubble period, and, accordingly, bubble departure frequency, and bubble footprint size distributions. This postprocessing requires several image segmentation techniques, whose details are extensively discussed in a previous communication[20]. In brief, the detection of nucleation sites is obtained as follows. A nucleation event produces a peak in the heat flux distribution, $q_w''(x,y,t)$. A dry spot grows underneath the bubble as the micro-layer evaporates around it, which leads to a local drop in the heat flux. Based on this observation, nucleation sites coincide with peaks in the measured probability scalar field, $p_n$, calculated as

$$p_n(x,y) = \sum_{f=1}^{N_f-1} \left[ \begin{array}{l} \left( \frac{d^2 q_w''(x,y,f)}{dx^2} < 0 \right) \\ \cdot \left( \frac{d^2 q_w''(x,y,f)}{dy^2} < 0 \right) \\ \cdot \left( \frac{d^2 q_w''(x,y,f+1)}{dx^2} > 0 \right) \\ \cdot \left( \frac{d^2 q_w''(x,y,f+1)}{dy^2} > 0 \right) \end{array} \right]. \tag{4}$$

where f is the frame index and $N_f$ is the number of frames (typically 5000 for 2 s). Note that the logical operators give 1 when the condition is satisfied and 0 if it is not. A sample probability map is shown in Fig. 4e. The nucleation sites are automatically detected as the peaks of the $p_n$ distribution. However, we always conduct a thorough visual check of the time-dependent temperature and heat flux distributions that may result in the removal or addition of nucleation sites. Once the nucleation sites are identified, the nucleation site density $N''$ is obtained as the ratio between the number of nucleation sites $N_s$ and the active heating area $A_h$. Then, the average bubble growth and wait time are identified for each of the nucleation sites, by analyzing the variation in temperature and heat flux (See Fig. 4e). For a given nucleation site, a nucleation event coincides with a sharp peak in the heat flux evolution (point 1 to 2 in the right figure), or a temperature drop (point 1 to 2 in the middle figure). As a dry spot forms underneath the bubble (after point 2), the heat flux drops suddenly, and the wall temperature starts increasing (points 2 through 3). As the bubble departs from its nucleation site, cold liquid quenches the site causing a second heat transfer peak associated with a wall temperature drop (point 4). Then, a new thermal boundary layer grows in the liquid until a new nucleation event occurs. The bubble period, $t_b$, is measured as the distance between two nucleation events (point 1 to 1'), the reciprocal of which gives the bubble departure frequency, $f$. The bubble growth time, $t_g$, is measured as the difference between the bubble period and the bubble wait time, $t_w$. Average quantities are obtained by tracking every nucleation event at each of the nucleation sites. They are calculated as

$$\alpha = \frac{1}{\sum_{s=1}^{N_s} N_{e,s}} \sum_{s=1}^{N_s} \sum_{n_{e,s}=1}^{N_{e,s}} \alpha_{n_{e,s}}. \tag{5}$$

where $\alpha$ is the variable of interest (e.g., $ft_g$), $N_s$ is the number of nucleation sites, and $N_{e,s}$ is the number of events for the s-th

nucleation site. The bubble footprint area distribution is determined by segmenting the images to distinguish the bubble footprint from the rest of the working fluid. The details of the image segmentation process are described in a previous communication[20]. This process also allows to separate discrete, non-interacting bubbles, i.e., bubbles that nucleate and depart from the surface without merging other bubbles. The footprint area distribution of non-interacting bubbles is found to be an exponential decay function[14]:

$$p(A) = ce^{-cA}. \tag{6}$$

where c is obtained by fitting the measured distribution. Assuming that the footprint of isolated bubbles is circular (i.e., $A = \pi r^2$), the corresponding equivalent radius distribution is given by

$$p(r) = 2\pi r c e^{-c\pi r^2}. \tag{7}$$

We can analytically demonstrate that the average value of this distribution, i.e., the average bubble footprint radius, $R$, is related to $c$:

$$R = \int_0^\infty r p(r) dr = \frac{1}{2\sqrt{c}}. \tag{8}$$

Thus, the bubble footprint radius distribution can be written as:

$$p(r) = \frac{\pi r}{2R^2} e^{-\frac{\pi r^2}{4R^2}}. \tag{9}$$

Equation (9) is used to sample the size of bubble footprint in the stochastic model, and it is uniquely defined once the average footprint radius $R$ is given.

**Boiling surfaces used for the water experiments.** We test seven different boiling surfaces:

- a plain ITO surface;
- a plain chromium (Cr) surface;
- a plain Alumina surface;
- a plain copper surface covered by copper oxide (CuO) nanoleaves;
- a plain titanium surface covered by zinc oxide (ZnO) nanowires;
- a plain ITO surface covered by a porous layer of silicon dioxide ($SiO_2$) nanoparticles;
- a rough zirconium (Zr) surface;

All surfaces are fabricated on the sapphire substrate described before. The ITO, Cr, and Alumina surfaces are deposited directly on the sapphire substrate by sputtering using a PRO Line PVD 75 by Kurt J. Lesker. Zinc oxide nanowires are created following the process described by Ko et al.[21]. Copper oxide nanoleaves are chemically etched following the same process as Rahman et al.[22]. The porous layer of silicon dioxide nanoparticles is coated using the layer-by-layer procedure developed by Lee et al.[23]. The rough Zr surface is fabricated according to the process described by Su et al.[15]. The ITO, Alumina, Cr and rough Zr surfaces have a contact angle with DI water of ~85°, 10°, 29° and 47° respectively (measured with an optical goniometer in air). However, all other surfaces (CuO nanoleaves, ZnO nanowires and $SiO_2$ nanoparticles) are super-hydrophilic. The apparent contact angle on these surfaces is 0, as a liquid droplet would be rapidly absorbed by the super-hydrophilic structures. The parameter used to characterize this super-hydrophilicity is the so-called wicking number. The wicking number, Wi, of these surfaces has been measured using the same definition and protocol proposed by Rahman et al.[24]. Fig. 5h summarizes the measured contact angle and wicking number, together with the surface roughness. Scanning Electron Microscope images of these surfaces are also shown in Fig. 5.

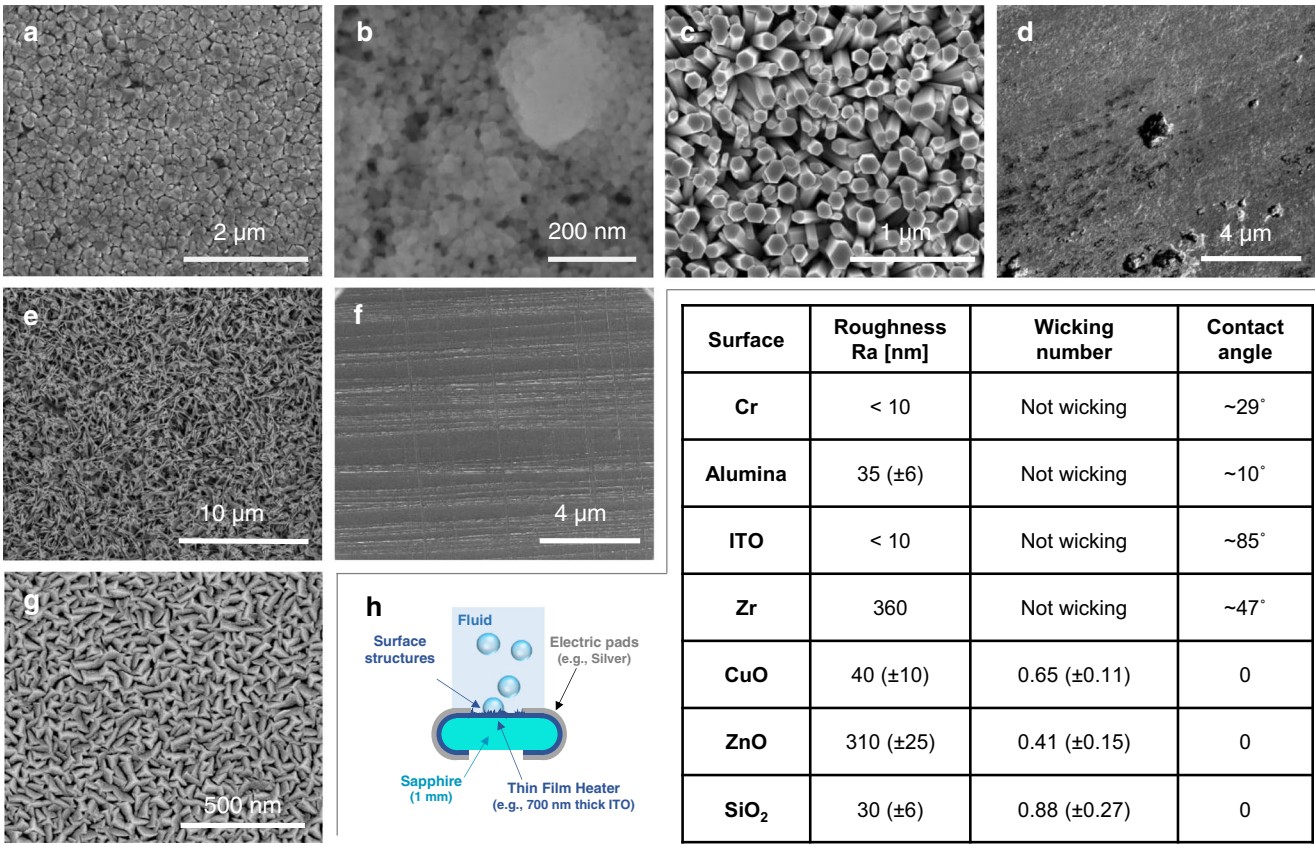

| Surface | Roughness Ra [nm] | Wicking number | Contact angle |
|---------|-------------------|----------------|---------------|
| **Cr** | < 10 | Not wicking | ~29˚ |
| **Alumina** | 35 (±6) | Not wicking | ~10˚ |
| **ITO** | < 10 | Not wicking | ~85˚ |
| **Zr** | 360 | Not wicking | ~47˚ |
| **CuO** | 40 (±10) | 0.65 (±0.11) | 0 |
| **ZnO** | 310 (±25) | 0.41 (±0.15) | 0 |
| **SiO$_2$** | 30 (±6) | 0.88 (±0.27) | 0 |

**Fig. 5 | SEM images of the surfaces tested with water and surface properties. a** ITO. **b** Silicon dioxide nanoparticles. **c** Zinc oxide nanowires. **d** Alumina. **e** Copper oxide nanoleaves. **f** Rough zirconium. **g** Chromium. **h** Schematic of the heater and measured roughness, wicking number, and contact angle for the tested surfaces.

**Experiments with liquid nitrogen.** The pool boiling experiment with liquid nitrogen is run with a specific setup, due to the peculiar behavior and boiling temperature of cryogenic fluids. Liquid nitrogen is boiled at the bottom of a thermally-insulated boiling cell by energizing a heating element (see Fig. 6a). The heater is the same as the ITO coated sapphire substrate used for the water tests. The boiling cell consists of a 7.6 cm side cube of aluminum, with a cartridge inserted on its bottom side and supporting the heater. The cartridge is made of low-thermal conductivity resin to reduce parasitic heat leaks. The boiling cell is filled with industrial-grade liquid nitrogen from a Dewar. The ITO is in direct contact with the liquid nitrogen and oriented horizontally upward. The cell is placed inside of a moisture-free chamber to prevent frosting on the optical windows and on the sapphire substrate. The chamber has tubing ports for filling with moisture-free nitrogen gas as well as liquid/gas nitrogen and electrical feedthroughs for the operation of the boiling cell. The nitrogen gas formed by boiling is directly evacuated to the atmosphere, and the boiling cell is regularly refilled with liquid nitrogen to keep the fluid level at 7 cm above the heating surface.

The cryogenic boiling experiment is performed at ambient pressure and saturated condition, same as the DI water experiments. At atmospheric pressure, the liquid nitrogen saturation temperature (i.e., −195.8 °C) is too low for measuring the temperature of the ITO using the infrared thermometry techniques. We also avoided the use of thermocouples due to the parasitic heat leak they can induce in the metallic sheath. Instead, we developed in-house a thin-film resistance thermometer (RTD). The RTD consists of a 500 nm-thick thin film of chromium oxide coated on the back side of the sapphire substrate (i.e., facing the moisture-free chamber). By tuning the content of oxygen inside the chromium oxide, the thin-film is made to behave as a semi-conductor, whose resistivity varies appreciably with its temperature. The RTD is driven by a 1 mA current generating a negligible heating power in the substrate (i.e., below 2 mW), and the voltage across the sensor is measured using a 4-point probe technique. The temperature of the RTD is obtained from the voltage across the RTD and by linear interpolation between 2 reference points. We use the saturation temperature of liquid nitrogen and liquid argon (i.e., −186.0 °C) as reference points. The thermal conductivity of sapphire reaches values as high as 1200 W/m/K at the boiling temperature of liquid nitrogen. With this thermal conductivity, we can safely assume that the substrate is isothermal and thus the temperature measured by the RTD is also the temperature of the boiling surface. The space-averaged heat flux to the fluid is simply given by the Joule heating superficial power density, $q''_h$. In order to capture the boiling dynamics and quantify the desired boiling parameters, the cryogenic pool boiling experiment is equipped with an optical setup allowing us to perform phase detection measurements. This technique allows tracking the phase (i.e., liquid and gaseous nitrogen) in contact with the heating surface (i.e., ITO) during boiling. The measurement method works on the principle of partial internal reflection of an incoherent quasi-monochromatic colored LED light at the ITO−nitrogen interface. With this technique we can track the bubble footprints and more generally, the position of the triple contact lines. This method was developed by Kossolapov et al. and applied to DI water[17]. Figure 6b shows a schematic of the optical setup as well as the typical raw image of the liquid nitrogen boiling process recorded by a high-speed video camera. A quasi-monochromatic red LED light beam is shined from the back side of the substrate with an incidence angle of 20° with the heating surface normal. The amount of light reflected depends on the index of refraction of the fluid in contact with the ITO. Gaseous nitrogen has an index of refraction much more different than liquid nitrogen from the index of refraction of sapphire.

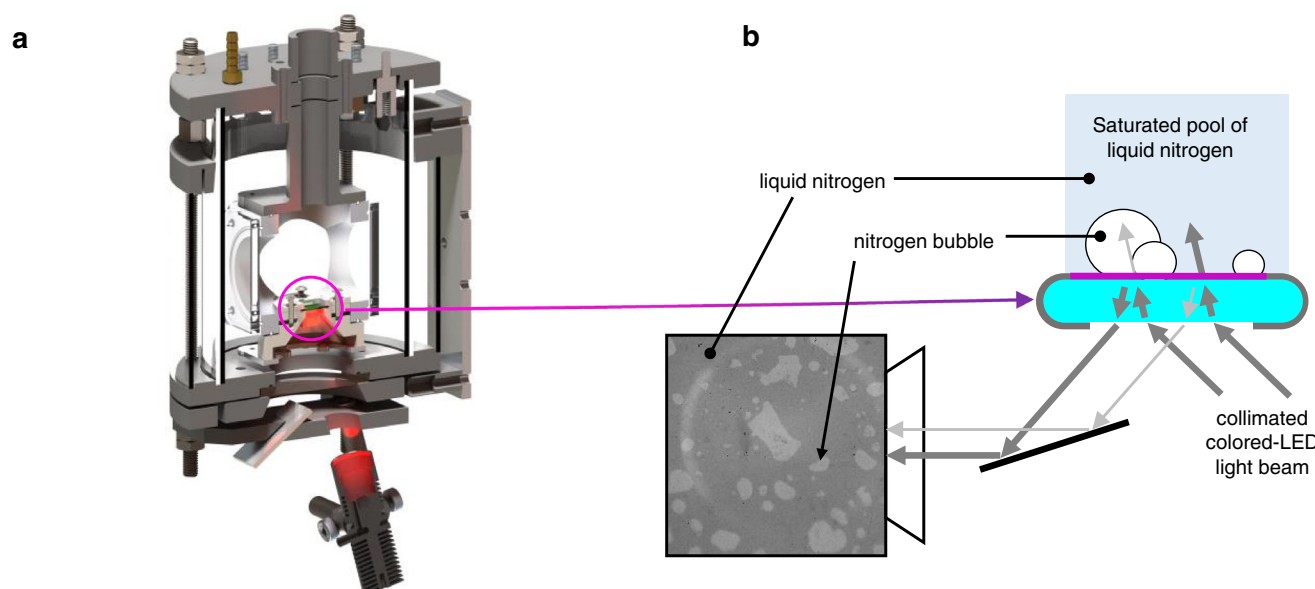

**Fig. 6 | Test sections used for cryogenic boiling experiments and phase detection technique. a** Cross sectional view of the pool boiling apparatus used for boiling experiments with liquid nitrogen. **b** Illustration of the phase-detection technique.

Thus, gaseous nitrogen reflects more light than liquid nitrogen. The reflected light is then captured by a high-speed video camera, producing contrasted gray images with bright and dark indicating gas and liquid nitrogen in contact with the heated surface, respectively. Our light source is a quasi-monochromatic LED with a spectrum centered around 620 nm. We use a high-speed video camera (Phantom v2512) with a temporal resolution of 30,000 frames per seconds and a pixel resolution of 5.4 μm. This is sufficient to resolve the length and time scales of the boiling process even with cryogenic fluids. Raw phase-detection images (such as the one shown in Fig. 6b) are post-processed to facilitate the measurements of the boiling parameters of interest. The post-processing of the raw images consists of a background removal followed by few steps of filtering to eliminate potential artefacts, such as dust on the optical setup. The boiling parameters of interest (i.e., nucleation site density $N''$, product of the nucleation frequency by the growth time $ft_g$, average radius $R$ of discrete bubbles, and overall bubble footprint area distributions) can be measured from the post-processed phase-detection recordings. The measurement of $R$ and $ft_g$ requires isolating each bubble in order to characterize them individually. The post-processed recordings are binarized and segmented, allowing us to identify each individual bubbles and all bubble clusters. The methodology used to evaluate each boiling parameter is briefly described hereafter. $N''$ is evaluated by tracking manually, on each recording, the position from where bubbles nucleate. The distinct positions, i.e., the active nucleation sites, are counted, and the total is divided by the surface area. Bubbles cannot be observed exactly at the moment of nucleation, but when they reach a sufficient size (typ., about 3–5 pixels). Therefore, 2 seemingly distinct positions of nucleation may be misinterpreted as separate nucleation sites. To avoid an overestimation of $N''$, 2 close-enough reported positions are considered as a single nucleation site. $R$ is evaluated by calculating the surface area of each bubble and clusters from the binarized recordings and for each frame. For high heat flux, typically close to CHF, very large clusters are excluded from the evaluation of $R$. The exclusion threshold is empirically defined by the cluster whose surface area is larger than the 99th percentile of the size of all clusters. For each nucleation site, the site-wise $ft_g$ is evaluated as the fraction of time that the nucleation site is covered by vapor. It is obtained by averaging in time all binarized images from the same recording (i.e., the same operating

conditions), where 0 and 1 indicate liquid and gaseous nitrogen, respectively. Then, $ft_g$ is simply calculated by averaging the site-wise $ft_g$ over all nucleation sites.

**Summary of the experimental results**. Figure 7 shows the boiling curves of the pool (a) and flow boiling (b) experiments. Time averaged heat flux and temperature are calculated by averaging in space and time the time-dependent distributions of temperature and heat flux measured with the infrared thermometry technique discussed before. In the case of liquid nitrogen, the heat flux is assumed equal to the heating power, and the boiling surface temperature is measured using a thin film RTD, as discussed in the previous section. Figure 7c shows the distribution of the measured bubble footprint area right before the boiling crisis, i.e., at the last operating heat flux for which we could achieve a stable nucleate boiling regime. Note that, no matter the boiling curve, these distributions are all power laws. Finally, Tables 1 and 2 report the values the values of heat flux, wall superheat, average bubble radius, $R$, product of bubble growth time and departure frequency, $ft_g$, and nucleation site density for all surfaces and operating conditions. We note that the value of the $ft_g$ product tends to increase for all surfaces and flow conditions with the heat flux. This observation is consistent with the observations of other scientists[25,26] (e.g., using sapphire-ITO heaters similar to the ones we used). Similarly, the value of the CHF measured in pool boiling on sapphire-ITO heaters is in the range of the values observed in literature[25–29].

The uncertainty on the heat flux is typically ±20 kW/m² and it is driven by the uncertainty on the voltage drop measurement. The uncertainty on the wall superheat, i.e., the wall temperature is within ±0.3 °C, and it is driven by the accuracy and precision of the thermocouples used for the calibration of the infrared signal. The uncertainty on the average bubble radius is within ±5 %, and it is dominated by the uncertainty on the image segmentation process. The uncertainty on the $ft_g$ product is dominated by the standard deviation of its distribution, and it is typically within ±10 %. The uncertainty on the nucleation side density is estimated within ±5%, based on a benchmark exercise comparing the measurements provided by three independent users. The effect of measurement uncertainties is shown in Fig. 7d, e. In this Figures, we plotted a 2D version of the 3D plots shown in Fig. 3b, c. On the *y*-axis we have the

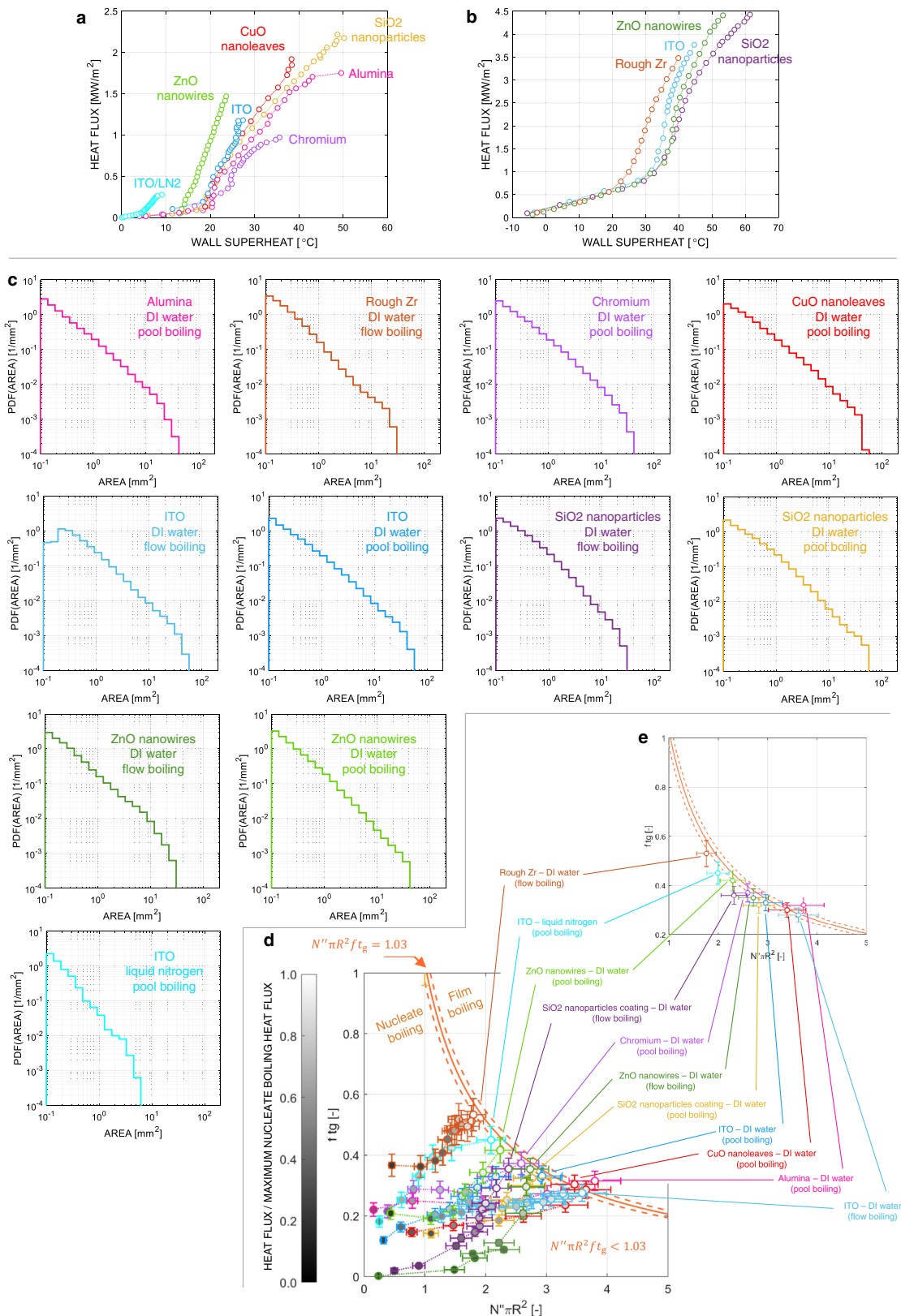

product $ft_g$, while on the x axis we have the product $N''\pi R^2$ (i.e., both axes are non-dimensional). Figure 7d reports all the boiling points (i.e., it is the equivalent of Fig. 3b), with error bars. Figure 7e only reports the points at the boiling crisis (i.e., it is equivalent to Fig. 3c). As shown, the uncertainty in the measurements does not change the conclusions drawn based on Fig. 3b, c.

## Stochastic model

We modeled the bubble interaction process using the stochastic model introduced in our previous work[14,15]. This model uses measured $N''$, $ft_g$, and $R$, as inputs and captures how the distribution of the bubble footprint area change with these parameters, i.e., with the operating conditions, from the onset of nucleate boiling till the

**Fig. 7 | Experimental results. a** Pool boiling curves. All experiments are run with DI water, except one, run with liquid nitrogen (LN2) (Source data are provided as a Source Data file). **b** Flow boiling curves. All experiments are run with DI water (Source data are provided as a Source Data file). **c** Distribution of the measured bubble footprint area right before the boiling crisis, i.e., at the last operating heat flux for which we could achieve a stable nucleate boiling regime. **d** Representation of the experimental boiling triplets $(N'', R, ft_g)$ for several boiling surfaces and operating conditions in a 2D plot with the product $ft_g$ on the $y$-axis and the product $N''\pi R^2$ on the x axis (Source data are provided as a Source Data file). The dot grayscale is proportional to the ratio between the heat flux and the maximum heat flux that each surface (represented by different line colors) can remove in nucleate boiling. White dots indicate the experimental boiling crisis and lie on the theoretical critical surface $N''\pi R^2 ft_g = 1.03$. **e** Simplified view of the critical surface (dashed line) and the experimental boiling triplets $(N'', R, ft_g)$ at the boiling crisis for all surfaces and operating conditions. Dashed lines represent the standard deviation on the theoretical critical constant.

boiling crisis. Given a surface of area $A_h$, we randomly create $A_h N''$ nucleation sites. Selected a nucleation site, the probability that a bubble is growing out of that site, as we observe it, is $ft_g$. Thus, given a random number between 0 and 1, if it is smaller than $ft_g$ we generate a bubble, otherwise, we move to another nucleation site. The radius of the bubble footprint, if any, is also generated randomly. It is sampled from the measured radius distribution, p($r$), of the bubbles that never interact with other bubbles throughout their life cycle (defined by the average footprint radius, $R$, through Eq. (9)). We repeat this process by looping over all nucleation sites in random order. If a site is already covered by a bubble footprint (i.e., the bubble generated by another nucleation site) we move to the next site. The typical result of one iteration is sketched in Fig. 8a. At the end of every iteration, we sample the size of the bubble patches, also called bubble clusters, and then we repeat the process from the beginning, i.e., re-starting with the creation of nucleation sites, as many times as necessary to achieve a converged distribution of the bubble clusters area. Sample predictions of this model are shown in Figs. 2e and 2f. As shown in these figures, the boiling crisis coincides with the occurrence of a power law distribution, and can be predicted based on the bifurcation of the bivariate probability p($A_G, A_{SG}$) of the giant and second giant bubble cluster areas (see Fig. 2f), $A_G$ and $A_{SG}$, respectively. Precisely, at each iteration, we sample $A_G$ and $A_{SG}$. Doing so, after many iterations we achieve a converged p($A_G, A_{SG}$) distribution. Examples of p($A_G, A_{SG}$) in different operating conditions are provided in Fig. 8b. As shown in the Figure, the peak of the distribution experience a sudden jump in correspondence of the critical conditions, for which we observe a power law distribution. Before critical conditions the peak values of $A_G$ and $A_{SG}$ are very similar, i.e., the bubble interaction is stable. Instead, at critical conditions, there is a bifurcation with the giant cluster area growing bigger and bigger, while the second giant bubble cluster disappears (see Fig. 2f). This instability in the bubble interaction process is the signature of the boiling crisis. We have used this model to determine critical combinations of $N''$, $R$, and $ft_g$, and found Eq. (1), i.e.,

$$N''\pi R^2 ft_g = C$$

as discussed in the main body of the paper. We observed that the values of $C$ depends on the non-dimensional boiling area $A_h/\pi R^2$. We run multiple simulations of over 30,000 iterations each for several values of the non-dimensional boiling area in order to obtain statistically converged results and determine the correlation between $C$ and $A_h/\pi R^2$. Figure 8c shows $C$ as a function of $A_h/\pi R^2$. As the non-dimensional area becomes infinitely large ($\sim 10^6$), the critical constant reaches an asymptotic value around 1.15. Interestingly, this value agrees rather well with the critical filling factor for an infinitely large system in the 2D continuum percolation with constant-size circles, i.e., 1.13[30]. This finding corroborates the idea that the boiling crisis falls into the class of first-order phase transition phenomena. Importantly, this sensitivity study reveals that, in the range of $A_h/\pi R^2$ typical of our experiments, i.e., 50 to 300, but also of other experiments and applications, the mean value of the parameter $C$ is very close to 1. Precisely, for a nondimensional area of 100, $C$ is equal to 1.03 and has a $\sigma$ of 0.06.

In addition to this sensitivity study to the nondimensional area, we also tested the robustness of this analysis with respect to our modeling assumptions. For instance, when we sample the bubble radius according to a normal distribution with the same average value and standard deviation as the exponentially decaying function, the critical constant for the same non-dimensional area does not change. Similarly, if instead of taking a unique, fixed value for $ft_g$, we use a distribution, e.g., power law distribution such that $p(ft_g) \propto (ft_g)^{-\gamma}$, the critical constant is almost the same. Precisely we obtain 1.07 and 1.10 for $\gamma$ equal to 1.5 and 2.5, respectively. However, the value of the critical constant remains within the range of its standard deviation.

### Reconciling percolation and hydrodynamic theory

Consider a pool of liquid in saturated conditions and a control volume as shown in Fig. 8d. The volume exchanges energy at the bottom surface (with a heat flux $q''_w$) and at the top boundary, by releasing bubbles and receiving saturated liquid.

In equilibrium conditions, the energy balance in the control volume leads to the following scaling law:

$$q''_w \propto \rho_v h_{lv} N'' \pi R_d^3 f. \tag{10}$$

where $R_d$ is the departure radius of bubbles, $N''$ is the nucleation site density, $f$ is the bubble departure frequency, $\rho_v$ is the vapor density, and $h_{lv}$ is the latent heat of vaporization. We may multiply and divide the right-hand side by $t_g$ and rearrange this equation as follows:

$$q''_w \propto \rho_v h_{lv} N'' \pi R_d^2 ft_g \frac{R_d}{t_g}. \tag{11}$$

We may assume that, at the boiling crisis, $R_d/t_g$ scales as the critical vapor velocity identified by Zuber[31]:

$$\left(\frac{R_d}{t_g}\right)^2 = (g\sigma(\rho_l - \rho_v))^{1/2} \frac{\rho_l + \rho_v}{\rho_l \rho_v}. \tag{12}$$

By substituting Eq. (12) into Eq. (11), and assuming that, everything else being the same, the bubble footprint radius scales with the bubble departure radius, i.e., $R \propto R_d$, Eq. (11) becomes

$$q''_w \propto \rho_v h_{lv} N'' \pi R^2 ft_g (g\sigma(\rho_l - \rho_v))^{\frac{1}{4}} \left(\frac{\rho_l + \rho_v}{\rho_l \rho_v}\right)^{\frac{1}{2}}. \tag{13}$$

Per our percolation criterion, at the boiling crisis we should have $N''\pi R^2 ft_g \sim 1$. With this assumption and after some mathematical manipulations, Eq. 13 becomes

$$q''_w \propto \rho_v h_{lv} \left(g\sigma \frac{(\rho_l - \rho_v)}{\rho_v^2}\right)^{\frac{1}{4}} \left(\frac{\rho_l + \rho_v}{\rho_l}\right)^{\frac{1}{2}}. \tag{14}$$

Which is Zuber's correlation for nucleate boiling CHF[4,31]. At ambient pressure, the liquid density is much higher than the vapor

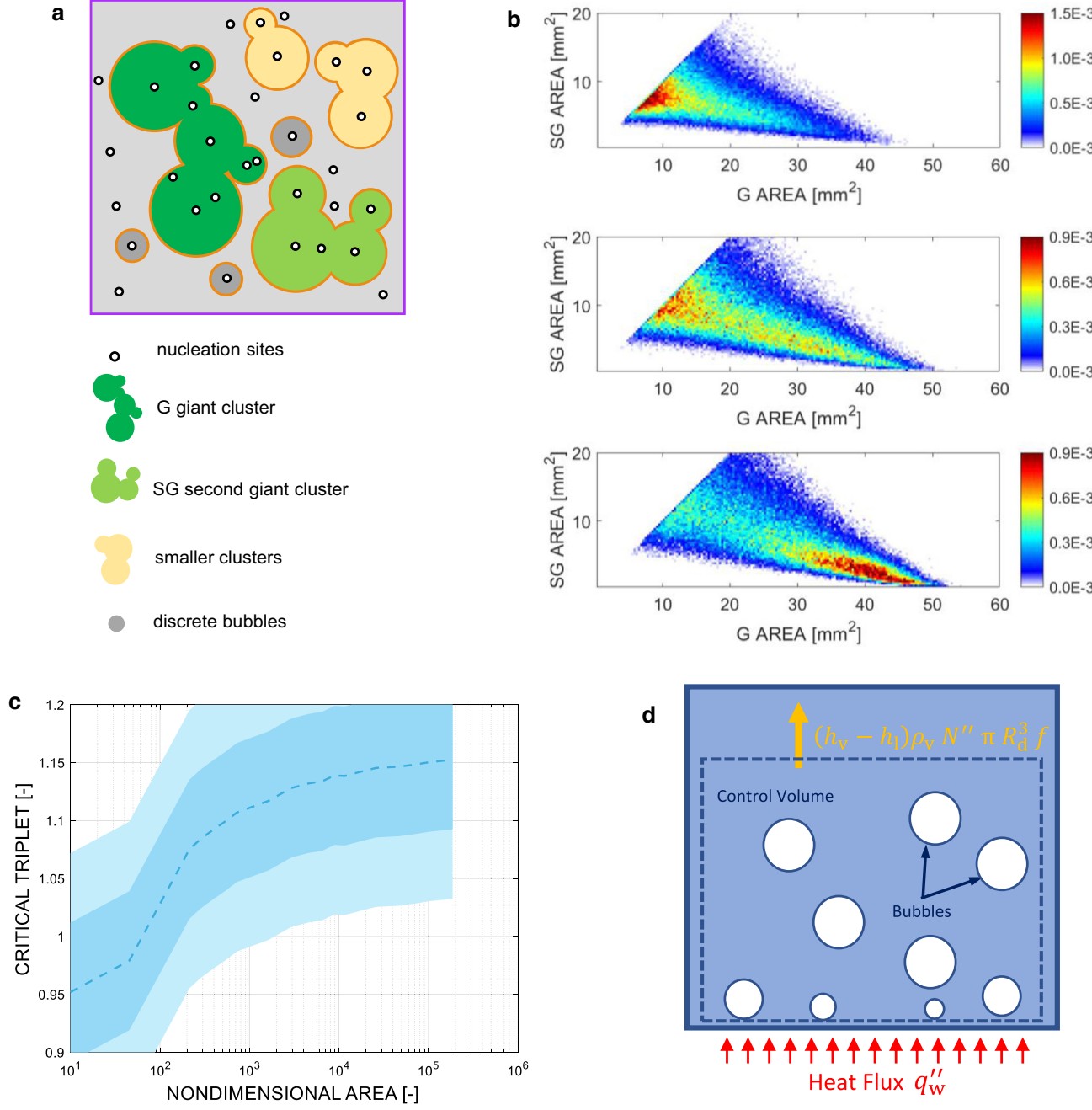

**Fig. 8 | Results of the stochastic bubble interaction model and control volume of hydrodynamic scaling analysis. a** Example of the stochastic model output for one iteration. One can recognize the presence of a giant G bubble cluster, a second giant SG bubble cluster, smaller bubble clusters and discrete, non-interacting bubbles. **b** Examples of bivariate distributions of the giant and second giant bubble cluster areas (in 1/mm⁴) for subcritical (top), critical (middle), and supercritical (bottom) conditions. **c** Sensitivity of the critical triplet to the nondimensional area. Critical triplet (dashed) line, $\pm\sigma$ (light blue), and $\pm2\sigma$ (teal) vs. nondimensional area $A_{\mathrm{h}}/\pi R^2$. **d** Control volume for the hydrodynamic scaling analysis. The volume exchanges energy at the bottom surface (with a heat flux $q''_{\mathrm{w}}$) and at the top boundary, by releasing bubbles and receiving saturated liquid.

density, thus

$$q''_{\mathrm{w}} \propto \rho_{\mathrm{v}} h_{\mathrm{lv}} \left( g\sigma \frac{(\rho_{\mathrm{l}} - \rho_{\mathrm{v}})}{\rho_{\mathrm{v}}^2} \right)^{\frac{1}{4}}. \tag{15}$$

which is known as Kutateladze-Zuber correlation. In summary, our percolation theory provides a scaling with the fluid properties and operating conditions consistent with the scaling identified by Kutateladze and Zuber.

## Data availability

Source data are provided in this paper. Other raw data are available from the corresponding author upon request. Source data are provided with this paper.

## Code availability

The computer script of the stochastic model that supports the findings of this study is available from the corresponding author upon request.

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

## Acknowledgements

We thank E. Baglietto, J. Buongiorno, and S. Yip for discussions. This material is based upon work supported by the U.S. Department of Energy contract No. DEAC05-00OR22725 (M.B.) and the National Science Foundation under award numbers 2019245 (M.B.) and 2018995 (M.M.R.).

## Author contributions

M.B. and M.M.R. planned and supervised the project. C.W., A.K., F.C., G.M.A. and B.P. developed the experimental setups and implemented the measurement techniques. C.W. M.M.R., J.H.S., G.S., F.C. conducted the experiments. L.Z., G.S., J.H.S. and C.W. developed the post-processing techniques and analyzed the experimental data. All authors contributed to the preparation of the manuscript.

## Competing interests

The authors declare no competing interest.
