## [Peer Review File · Nature Communications]

REVIEWER COMMENTS

Reviewer #2 (Remarks to the Author):

In my opinion, the paper has been considerably improved further to the comments of all reviewers. In particular, all my personal comments were addressed more than satisfactorily.

I am not yet fully convinced that this theory represents the final answer to a long and historical debate about the causes of critical heat flux in boiling. However it is for sure an original and fresh contribution, that is worth to be published and discussed among the scientific community.

Reviewer's Comments

I thank the authors for responding to my earlier comments. Again, I emphasize that the authors have proposed a hypothesis, developed a model for critical heat flux and validated the predictions with their own data. What they propose is not a law as there are so many exceptions. In response to my earlier comments the authors themselves acknowledge that operating conditions affect the three boiling parameters (and in turn the predicted critical heat flux). The operating conditions can be size of the surface, morphology of the surface and orientation of the surface in relation to gravitational vector. As such, I am unable to accept what the authors claim to be a law! They have simply used their own data to prove a point. I suggest that the authors change the title of the paper to: "A unifying model for the boiling crisis".

There are several other issues with their development of the model:

- (i) In response to my earlier comment (iii) the authors report that the ratio of the wait time to the growth time decreases from 5.6 to 2.5 as heat flux was increased 14 times. This is not consistent with the many data reported in the literature. Waiting time depends among other variables on the spatial and temporal variation of heat transfer on the fluid side, on the thickness of the substrate and the thermophysical properties of the substrate. In fact, waiting time can be smaller than the growth period.
- (ii) With respect to my earlier comment (iv) critical heat flux for a contact angle of 85° has been reported to be half of that for a contact angle of about zero. See the work of Winterton on a horizontal surface and that of Liaw for a vertical surface.
- (iii) The authors do not discuss the merger of bubbles normal to the surface. In fact, this can be the main mode responsible for critical heat flux on well wetted horizontal surfaces.
- (iv) Critical heat flux on a horizontal surface can be affected not only by the contact angle, the size of the surface and the size of the surrounding surface also influence it.

- (v) Recently I saw a paper published in the Journal of Electronic Packaging Dec. 2021, by She and Dhir. The authors clearly show the existence of vapor columns near the critical heat flux, and also the effect of contact angle, size of the heater and size of the surrounding vessel. How would a frequency be defined when bubbles at a nucleation site merge in the vertical direction?

Finally, I am not convinced that the authors have developed a universal law when they only use their own data to validate their model. I will agree to acceptance of the paper provided the title of the paper is changed.

Referee #1 (Remarks to the Author):

I thank the authors for responding to my earlier comments. Again, I emphasize that the authors have proposed a hypothesis, developed a model for critical heat flux and validated the predictions with their own data. What they propose is not a law as there are so many exceptions. In response to my earlier comments the authors themselves acknowledge that operating conditions affect the three boiling parameters (and in turn the predicted critical heat flux). The operating conditions can be size of the surface, morphology of the surface and orientation of the surface in relation to gravitational vector. As such, I am unable to accept what the authors claim to be a law! They have simply used their own data to prove a point. I suggest that the authors change the title of the paper to: "A unifying model for the boiling crisis".

Dear Referee #1,

We are thankful for your careful and thoughtful review of our revised paper. We answer your specific questions and comments below using RED fonts. Changes to the manuscript motivated by your questions and comments are also written in RED fonts.

It seems that the major concern relates to the choice of the word "law" in the title and throughout the manuscript, and we wish to address this comment immediately hereafter.

After careful consideration, we think that a better word to describe what we achieved is "criterion." We prefer to use "criterion" rather than "model" because we believe that our work goes beyond the model, as we also obtain (using the model) a non-dimensional scaling criterion to determine whether boiling is stable (i.e., $N'' \pi R^2 f t_g < C$) or a boiling crisis may occur (i.e., $N'' \pi R^2 f t_g = C$), and support this criterion with experimental results. Thus, we feel that the word "criterion" is more appropriate than "model".

There are several other issues with their development of the model:

- (i) In response to my earlier comment (iii) the authors report that the ratio of the wait time to the growth time decreases from 5.6 to 2.5 as heat flux was increased 14 times. This is not consistent with the many data reported in the literature. Waiting time depends among other variables on the spatial and temporal variation of heat transfer on the fluid side, on the thickness of the substrate and the thermophysical properties of the substrate. In fact, waiting time can be smaller than the growth period.

We completely agree with your comment.

In fact, we emphasize that our dataset covers a much broader wait time to growth time range compared to the mentioned above (see Extended Data Table 2 and 3 of the manuscript). In particular, at the boiling crisis, our $f t_g$ values range between 0.28 and 0.53. A value of 0.28 means that the wait time is approximately 2.6 times longer than the growth time. **However, a value of 0.53 means that ration between the wait time and growth time is 0.88, i.e., the wait time is smaller than the growth time, which agrees with your statement.** Note that, an $f t_g$ smaller than 0.5 indicates that the wait time is longer than the growth time. Vice versa, if $f t_g$ is larger than 0.5, the growth time is longer than the wait time.

We believe the confusion arose from the fact that in our answer to question (iii) of your earlier comments (i.e., "It can be deduced from the data that for DI water and ITO the ratio of waiting time to growth time is about 5 at half of the critical heat flux and 2 near critical heat flux. These ratios are not supported by numerical or experimental data reported in the literature."), our focus was to support our data with data obtained in the same conditions by other scientists in the conditions you mentioned, i.e., water and ITO, for which the available data are, to the best of our knowledge, the pool boiling data of Gerardi [1,2], as discussed in our earlier rebuttal. Clearly, for other surfaces and operating conditions the wait time to growth time ratios can be different, as supported by our data.

We hope that this clarification addresses your follow-up comment. As these data were already in the manuscript, no change was necessary. However, we added a section to clearly state the range of our experimental data.

[1] Gerardi, C. Investigation of the pool boiling heat transfer enhancement of nano-engineered fluids by means of high-speed infrared thermography. (Massachusetts Institute of Technology, 2009).

[2] Gerardi, C., Buongiorno, J., Hu, L. & McKrell, T. Infrared thermometry study of nanofluid pool boiling phenomena. *Nanoscale Res Lett* **6**, 1–17 (2011).

- (ii) With respect to my earlier comment (iv) critical heat flux for a contact angle of 85° has been reported to be half of that for a contact angle of about zero. See the work of Winterton on a horizontal surface and that of Liaw for a vertical surface.

We emphasize that, in response to your earlier comment (iv) (i.e., “The critical heat flux reported for ITO and deionized water for a contact angle of 85 degrees is much higher than reported in the literature.”), we provided evidence that the CHF values we measured with water on ITO are indeed consistent with those reported in literature on similar surfaces.

However, we believe that surface wettability is not a unique indicator of the CHF limit on the boiling surface. Surface morphology plays an equally important, if not more important role.

In support of this statement, we wish to cite the work of O’ Hanley et al. [3]. They have conducted a separate effect study of pool boiling CHF where they were able to change the surface wettability without changing the surface morphology to show that the CHF limit does not change for contact angles ranging between 112 and 0 degrees.

We may also reference a recent work from our group [4], aimed at elucidating the separate effect of wettability and surface changes (promoted by oxidation). By oxidizing a zircaloy surface, we have been able to obtain surfaces with different wettability without changing the surface morphology (we did surface characterization on the same exact spot of the boiling surface before and after oxidation). We have ascertained that changes of wettability from 50 to 30 degrees do not affect the boiling curves or the CHF. CHF only changes when the oxidation process modifies structurally the surface (and so the change in contact angle is a combination of a change in the intrinsic wettability and the surface morphology) [4].

We noticed that in the work of Winterton [5], the change of contact angle are attributed to oxidation. As the surface is aluminum, it is likely that the oxidation has created morphological changes to the surface. Similarly, Liaw and Dhir [6] explored the effect of contact angle by oxidizing copper surfaces, which are also subject to morphological changes when oxidized.

In summary, while we do not question the results of previous works, we feel that it is incautious to compare the boiling performance of two morphologically different surfaces based on the contact angle only. We also wish to re-emphasize that our CHF results, as discussed in response to your earlier comment, are in close agreement with CHF results from literature obtained in the same experimental conditions.

[3] O’Hanley, H., Coyle, C., Buongiorno, J., McKrell, T., Hu, L.W., Rubner, M. and Cohen, R., 2013. Separate effects of surface roughness, wettability, and porosity on the boiling critical heat flux. *Applied Physics Letters*, 103(2), p.024102.

[4] Seong, J.H., Wang, C., Phillips, B. and Bucci, M., 2022. Separate effect of oxidation on the subcooled flow boiling performance of Zircaloy-4 at atmospheric pressure. *International Journal of Heat and Mass Transfer*, 188, p.122620.

[5] Liu, Z. and Winterton, R.H.S., 1991. A general correlation for saturated and subcooled flow boiling in tubes and annuli, based on a nucleate pool boiling equation. *International journal of heat and mass transfer*, 34(11), pp.2759-2766.

[6] Liaw, S.P. and Dhir, D.V.K., 1986. Effect of surface wettability on transition boiling heat transfer from a vertical surface. In *International Heat Transfer Conference Digital Library*. Begel House Inc.

- (iii) The authors do not discuss the merger of bubbles normal to the surface. In fact, this can be the main mode responsible for critical heat flux on well wetted horizontal surfaces.

This point is very controversial. In fact, there is no consensus in the community. We tried to capture the essence of this controversy in the introduction of our paper:

“The boiling crisis is a century-old scientific problem. For years, it has been viewed as the outcome of a hydrodynamic instability occurring far from the heated surface^{Error! Reference source not found.}. When ... which eventually dries out. Antithetical descriptions consider the boiling crisis as a near-wall instability related to the characteristic of the nucleate boiling process at the heated surface. For instance, ...”

An experimental work supporting the near-wall nature of the boiling crisis by Gong et al. [7] has shown that in pool boiling (with an horizontal surface) the CHF limit does not depend on the height of the liquid pool, down to liquid heights in the order of a millimeter, which is the order of magnitude of the bubble footprint radius and departure diameters (with 1 mm tall liquid pools there are no Kelvin-Helmholtz instabilities in the liquid far from the wall, nor merging of bubbles normal to the surface).

Crucially, we have repeated the same experiments in our pool boiling apparatus and made a similar observation.

However, while we think we should keep an open mind with respect to this important question, our results seem to support the idea that the boiling crisis in nucleate boiling is triggered by near-wall instabilities, rather than far-field hydrodynamic effects (as we only used information related to the dynamic of bubbles at the heating surface to predict the boiling crisis). This is noteworthy, as our model also yields the same scaling of the critical heat flux (CHF) with the physical properties of the fluid and operating conditions (as shown in the paper) predicted by the Kutateladze-Zuber correlation, i.e., a far field model. Briefly, our idea seems to reconcile far-field and near-wall views of the boiling crisis. Motivated by your comment, we have added a short sentence to clarify this aspect in the revised manuscript.

[7] Gong, S., Ma, W. and Gu, H., 2014. An experimental investigation on bubble dynamics and boiling crisis in liquid films. *International Journal of Heat and Mass Transfer*, 79, pp.694-703.

- (iv) Critical heat flux on a horizontal surface can be affected not only by the contact angle, the size of the surface and the size of the surrounding surface also influence it.

We agree that the CHF limit (i.e., the maximum heat flux that can be removed in nucleate boiling in W/m^2) depends on a plethora of parameters such contact angles, surface size, size of the surrounding surface, and many more. However, as mentioned in our initial rebuttal, we emphasize that our goal is not to understand how a surface or operating conditions (i.e., the blue boxes in the figure below) modify our three boiling parameters (i.e., the orange boxes), but to reveal that the boiling crisis (the green box) is the result of the same stochastic process, which can be predicted from the three parameters using a unique criterion, i.e., Eq. 1 ($N'' \pi R^2 f t_g = C$), no matter the surface morphology and wettability, the fluid and operating conditions, the size of the boiling surface and the size of the surrounding surface.

The size of the surface and the size of the surrounding surface, and all the other parameter may directly and indirectly affect the bubble dynamics, provide different flow recirculation conditions, bubble detaching forces, convective cooling effects, and so on. However, the process leading to the boiling crisis should be the same, i.e., the same criterion applies to small surfaces.

For the sake of clarity (as mentioned in our paper and discussed in the **Stochastic Model** description of the **Methods** section), we emphasize that the percolation threshold is slightly dependent on the nondimensional boiling area* (for small values of the latter).

* As written in the paper, the nondimensional boiling area is defined as $A_h/\pi R^2$ where A_h is the area of the boiling surface and R is the average bubble footprint radius. In our analysis we assumed that the surface is not oblate, i.e., it does not have a principal direction.

- (v) Recently I saw a paper published in the Journal of Electronic Packaging Dec. 2021, by She and Dhir. The authors clearly show the existence of vapor columns near the critical heat flux, and also the effect of contact angle, size of the heater and size of the surrounding vessel. How would a frequency be defined when bubbles at a nucleation site merge in the vertical direction?

The pictures in the paper of She and Dhir [8] show indeed the presence of a vapor column far from the surface, which is typical. Also, from what we can see (we tried to zoom in Figs. 6, 9 and 11 of the paper), we think that the dark shadows corresponding to the boiling surface may indicate that individual bubbles still nucleate under the vapor column). However, we are afraid that these pictures do not shed any light on what is happening on the surface itself (as the space and time resolution of the images is too coarse, and there is no direct optical access to the boiling surface). It is impossible to say if there is bubble nucleation and detachment from the surface (with merging of the detached bubble with the hovering bubble column) or there are tiny, permanently dry spots connected to the hovering vapor column with vapor stems. However, we can exclude that the base of the large vapor column is completely and constantly dry (as this would be a post-CHF heat transfer regime!).

In brief, there are two possible answers to your question (i.e., “How would a frequency be defined when bubbles at a nucleation site merge in the vertical direction?”):

- If bubbles detach from the surface to merge with the hovering vapor column, their growth time is still defined as the time from nucleation until the moment they are completely detached, and the frequency is still defined as the inverse of the bubble period (which is the sum of growth and wait time).
- If there are tiny, permanently dry spots connected to the hovering vapor column, the product $f t_g$ is simply equal to 1 (but the criterion still holds!). Note that what we really need to know is $f t_g$ (not the frequency), i.e., the probability that a site is sustaining a bubble (or a dry spot).

We wish to emphasize that one could elucidate these aspects by using imaging techniques similar to ours (for example our IR measurements enable direct optical access to the boiling surface and are taken at 2000 frames per second and 100 microns per pixel). From what we can see from our side view of the experiment, our pool boiling process does not look different from the images shown by She and Dhir [8]. However, the infrared

imaging of the boiling surface reveals a much more complicated process, which cannot be captured with side views.

The trends observed by She and Dhir [8] make perfect sense. However, we humbly submit that advanced measurement techniques enabling direct optical access to the boiling surface and a very high temporal and spatial resolution are necessary to capture and understand the dynamic of the boiling process and the boiling crisis.

[8] She, Z. and Dhir, V.K., 2021. Parametric effects of heater size, contact angle, and surrounding vessel size on pool boiling critical heat flux from horizontal surfaces. *Journal of Electronic Packaging*, 143(4).

Finally, I am not convinced that the authors have developed a universal law when they only use their own data to validate their model. I will agree to acceptance of the paper provided the title of the paper is changed.

We wish to clarify, as also mentioned in the previous rebuttal, that we never made the claim that our criterion is “universal” but “unifying.” We believe we have already clarified the meaning of “unifying” in our R1 version of the paper, e.g., *“We emphasize that, in this context, the word **unifying** is used because this law combines and captures the synergistic and intertwined effect of the length and time scales involved in the boiling process, i.e., nucleation site density, bubble size, growth time and departure frequency. ...”*

We have already addressed the comment related to the title and the use of the word “law” at the beginning of the rebuttal. As discussed, we wish to title our paper “A unifying criterion of the boiling crisis.” We trust that you will find our proposal appropriate. Accordingly, the word “law” has been changed with “criterion” throughout the manuscript.

Referee #2 (Remarks to the Author):

In my opinion, the paper has been considerably improved further to the comments of all reviewers. In particular, all my personal comments were addressed more than satisfactorily.

I am not yet fully convinced that this theory represents the final answer to a long and historical debate about the causes of critical heat flux in boiling. However it is for sure an original and fresh contribution, that is worth to be published and discussed among the scientific community.

Dear **Referee #2**,

We appreciate your positive feedback. The boiling crisis has been indeed the subject of a long debate (in fact, more than on hundred years old). We are glad to shake this debate with (as you say) “an original and fresh idea” and first-of-a-kind data revealing new physical mechanisms. We are excited to share our findings with the community and see other scientists to critically evaluate and build upon our work.

REVIEWERS' COMMENTS

Reviewer #1 (Remarks to the Author):

With the change of the title of the paper and related text my main concern has been alleviated. However, the authors keep missing the point that in pool boiling of inviscid fluids it is the balance of buoyancy and surface tension forces that determines the maximum rate of vapor removal from the surface. From their work it is not clear under what physical situations it is the far field or the near field instability determines the outcome. The discussion of the authors on the effect of contact angle, the size of the heater and the size of the surrounding vessel and the contact angle of the surface surrounding the heater is lacking. At this stage I agree to support the publication of the paper in its revised form.

Referee #1 (Remarks to the Author):

With the change of the title of the paper and related text my main concern has been alleviated. However, the authors keep missing the point that in pool boiling of inviscid fluids it is the balance of buoyancy and surface tension forces that determines the maximum rate of vapor removal from the surface. From their work it is not clear under what physical situations it is the far field or the near field instability determines the outcome. The discussion of the authors on the effect of contact angle, the size of the heater and the size of the surrounding vessel and the contact angle of the surface surrounding the heater is lacking. At this stage I agree to support the publication of the paper in its revised form.

Dear Referee #1,

We appreciate your final recommendation.

We understand your comment about the role of buoyancy and surface tension for pool boiling conditions. However, we think that the role of these forces is accounted for through the fundamental parameters describing the bubble dynamics (i.e., bubbler radius, growth time, departure frequency, and nucleation site density). We emphasize that our model aims at capturing the connection between these fundamental parameters and the boiling crisis.

Similarly, as already mentioned in our previous rebuttal, we agree that the CHF limit (i.e., the maximum heat flux that can be removed in nucleate boiling in W/m^2) depends on a plethora of parameters such contact angles, surface size, size of the surrounding surface, and many more. However, we believe that the role of these parameters is to affect the bubble dynamics and, consequently, the boiling crisis. Same as before, we emphasize that our model aims at capturing the connection between the parameters describing the bubble dynamics and the boiling crisis.

We also wish to mention that we discuss the effect of contact angle in our paper. Precisely, we wrote: *“We emphasize that, while we did not test hydrophobic surfaces with static contact angles much larger than 90° , our paradigm contemplates all surface conditions. For instance, on hydrophobic surface, depending on nucleation sites size distribution and nucleation temperature, the boiling crisis may occur with a large number of small bubbles (i.e., with a small footprint) and a short wait time (i.e., a high ft_g)²³. In some other cases, the bubble departure diameter on hydrophobic surfaces may be significantly higher than on hydrophilic surfaces²⁴, and the boiling crisis may happen with a combination of large bubble footprint radius and smaller nucleation site density (and likely higher ft_g product). However, while experiments on superhydrophobic surfaces may be run in the future, the critical condition expressed by Equation (1) should not change.”*

Based on your comment, we modified this sentence to mention possible other effects (e.g., such as the size effects discussed in the Methods section), as follows:

“We emphasize that, while we did not test hydrophobic surfaces with static contact angles much larger than 90° , our paradigm contemplates all surface conditions and, possibly, heater geometry effects. For instance, on hydrophobic surface, ... combination of large bubble footprint radius and smaller nucleation site density (and likely higher ft_g product). However, while experiments on superhydrophobic surfaces and considering other effects (e.g., heater size and lateral confinement) may be run in the future, the critical condition expressed by Equation (1) should not change.”

Overall, the fact that the critical behavior can be captured using near-wall parameters for surfaces with different boiling behavior (i.e., nucleation site density, bubble growth time and departure frequency, and average footprint radius) and different CHF limits under the same or different operating conditions is in our view, a strong indication that the mechanisms triggering the boiling crisis can be found in the near-wall region. This is the reason why we wrote that our work “corroborates the hypothesis that the boiling crisis is a near-wall phenomenon.”

We are truly grateful for your constructive criticism throughout the rebuttal process. We are excited to share our findings with the community and see other scientists to critically evaluate and build upon our work.